# Synergy between in situ and altimetry data to observe and study the Northern Current variations (NW Mediterranean Sea).

**Alice Carret[(1)], Florence Birol[(1)], Claude Estournel[(2)], Bruno Zakardjian[(3)] and Pierre Testor[(4)]**

(1) LEGOS, Université de Toulouse-CNES-CNRS-IRD, OMP, 14 Av. E. Belin, 31400 Toulouse, France
(2) L.A. Université de Toulouse-CNRS, OMP, 14 Av. E. Belin, 31400 Toulouse, France
(3) Université de Toulon, CNRS/INSU, IRD, Mediterranean Institute of Oceanography (MIO), UM 110, 83957 La Garde, France
(4) LOCEAN, Sorbonne Université-CNRS-IRD-MNHN, Paris, France

## Abstract

During the last 15 years, substantial progress has been achieved in altimetry data processing, providing now data with enough accuracy to illustrate the potential of these observations for coastal applications. In parallel, new altimetry techniques improve the data quality by reducing the land contamination and by enhancing the signal-to-noise ratio. Satellite altimetry provides more robust and accurate measurements ever closer to the coast and resolve shorter ocean signals. An important issue is now to learn how to use altimetry data in conjunction with the other coastal observing techniques.

Here, we cross-compare and combine the coastal currents provided by large data sets of ship-mounted ADCPs, gliders, HF radars and altimetry. We analyze how the different available observing techniques, with a particular focus on altimetry, capture the Northern current variability at different time-scales. We also study the coherence/divergence/complementarity of the information derived from the different instruments considered. Two generation of altimetry missions and both 1-Hz and high-rate measurements are used: Jason 2 (nadir Ku-band radar) and SARAL/AltiKa (nadir Ka-band altimetry); their performances are compared.

In terms of mean speed of the Northern Current, a very good spatial continuity and coherence is observed at regional scale, showing the complementarity between all the types of current measurements. In terms of current variability, there is still a good spatial coherence but the Northern Current amplitude derived from altimetry, glider, ADCP and HF radar data differ, mainly because of differences in their respective spatial and temporal resolutions. If we consider the seasonal variations, 1-Hz altimetry captures ~60% and ~55% of the continental slope current amplitude observed by the gliders and by the ADCPs, respectively. For individual dates this number varies a lot as a function of the characteristics of the Northern Current at the corresponding date, with no clear seasonal tendency observed. Compared to Jason 2, the SARAL altimeter data tend to give estimations of the NC characteristics that are closer to in situ data in a number of cases. The much

noisier high-rate altimetry data appear to be more difficult to analyze but they provide current estimates that are generally closer to the other types of current measurements. Thus, satellite altimetry provides a synoptic view of the Northern Current circulation system and variability which helps to interpret the other observations. Its regular sampling allows the observation of many features that may be missed by irregular in situ data.

## 1. Introduction

Radar altimeters allow to estimate the sea surface height (SSH) variations along the satellite tracks at regular interval time. Providing a large number of continuous and accurate observations of the global oceans since more than 25 years, they have progressively evolved into one of the fundamental instruments for many scientific and operational oceanographic applications (Morrow and Le Traon, 2012). The SARAL mission and its first AltiKa Ka-band frequency radar, launched in 2013, has still improved the performance of satellite altimetry (Bonnefond et al., 2018). With the launch of Sentinel-3A, B, in February 2016 and in April 2018, the altimetry constellation was completed by the first instruments always operated at high resolution mode (commonly called Synthetic Aperture Radar or SAR). These new altimeters provide enhanced along-track resolution and reduced noise, in comparison to the conventional nadir-looking pulse limited Ku-band instruments used since the beginning of the altimetry era. In 2021, the SWOT mission, with its SAR interferometer in Ka-band measuring SSH over 120-km wide swaths, will be a new step forward (https://swot.jpl.nasa.gov/docs/SWOT_D-79084_v10Y_FINAL_REVA__06082017.pdf).

In coastal ocean areas, it is particularly important to monitor the sea level variations. About 10% of the world population lives in low-elevation coastal zones (Nicholls and Cazenave, 2010) exposed to hazards such as extreme events, flooding, shoreline erosion and retreat. The latter are expected to increase due to the combined effects of sea level rise, climate change, and increasing human activities. In coastal regions in particular, we expect a lot of advances from modern altimetry instruments and processing techniques. Indeed, conventional satellite altimetry missions have not been designed for the observation of the coastal dynamics. The strongest limitation is the modification of the radar echo in the vicinity of land but the sea level estimations derived are also impacted by inhomogeneity in the water surface observed by the radar and by less accurate corrections. Coastal altimetry measurements are much more difficult to interpret than in the open ocean and need a dedicated processing and specific corrections (Gommenginger et al., 2011; Cipollini et al., 2017a). The data resolution is also too low to capture the fine scales of the coastal ocean dynamics. As a consequence, most altimetry data collected in coastal zones over the last 25 years have been discarded in altimetry products and/or poorly exploited. A lot of efforts has been done during the last 15 years in the altimetry community to overcome these difficulties and substantial progress has been achieved on the data processing side (Roblou et al., 2011; Passaro et al., 2014; Valladeau et al., 2015; Cipollini et al., 2017a), starting to provide data with enough accuracy to illustrate the potential of altimetry for coastal applications (Passaro et al., 2016; Birol et al., 2017; Morrow et al., 2017). Moreover, the use of new altimetry techniques provides more robust and accurate measurements closer to the coast and allows to resolve shorter spatial scales (Dufau et al., 2016; Morrow et al., 2017). As an example, from Birol and Niño (2015), closer than 10 km to the coastline, available SARAL data is still ~60% and only ~31% for Jason-2. From Morrow et al. (2017), in summer, SARAL can detect ocean scales down to 35 km wavelength, whereas the higher noise from Jason-2 blocks the observation of scales less than 50-55 km wavelength. As a result, the capability of altimetry for the monitoring of coastal ocean dynamics has already been illustrated in a number of studies. Most of them concern shelf and boundary currents (Bouffard et al., 2008; Birol

et. al., 2010; Herbert et al., 2011; Jébri et al. 2016, 2017). Some others are related to sea level applications (Cipollini et al., 2017a). A more complete review of coastal altimetry applications can be found in Cipollini et al (2017b) and we can easily predict that the use of this instrument in coastal studies will be largely extended in the next years.

Today, observations used in coastal applications are mainly based on in situ instruments and satellite imagery (sea surface temperature and ocean color images). In order to answer to the need for monitoring of the coastal ocean environment, in situ observing systems gather informations in a growing number of regions such as along the Australian or US coasts (http://imos.org.au/; https://portal.secoora.org/; http://www.nanoos.org/; see also Liu et al., 2015). The different

techniques are often used in synergy, measuring different ocean state parameters on different time and spatial scales. Compared to altimetry, their spatial and/or temporal resolution is much more adapted to detect the coastal ocean variability. Nonetheless, in situ observations cover more limited areas and often provide time series that present large gaps which may be several days (buoy data, HF radars) to several months (glider, ship data). Moreover, optical imagery is often impacted by

clouds and does not provide any direct information on the changes occurring in the water column. The large advantage of satellite altimetry, and the reason of its success in the deep ocean, is that it offers almost-global and synoptic observations of the sea level, a geophysical parameter which can be related to the ocean circulation and many other dynamical features (eddies, waves, sea water changes, ...). An important issue is now to learn how to use altimetric data in synergy with the other

coastal observing techniques.

To study the contribution of altimetry amongst other types of coastal ocean measurements, the North-Western Mediterranean Sea (NWMed) represents a laboratory area. First, with a Rossby radius of only ~10 km, the region is associated to a variety of mesoscale and sub-mesoscale dynamical signals (see below). As a result it represents a challenge for altimetry. Secondly, the

number of in situ observations is relatively important in this region, allowing comparison with independent data. In the NWMed, the main feature of the surface ocean circulation is the Northern Current (called NC hereafter) which is formed in the Ligurian Sea (Taupier-Letage and Millot, 1986) and flows cyclonically along the Italian, French and Spanish coasts. This current presents a marked seasonal variability, with a maximum amplitude from February to April (Sammari et al.,

1995; Millot, 1991), and it meanders in a vast range of wavelengths (10-100 km). The mesoscale variability is higher in autumn and winter because of the larger baroclinic instability associated to strong and cold winds (Alberola et al., 1995; Millot, 1991). During the last 10 years, the NC has been intensively monitored by a variety of in situ data (moorings, research vessels, gliders and HF radars) collected from the MOOSE (Mediterranean Ocean Observing System for the Environment)

integrated observing system. Despite a width of only 30-50 km, through the comparison with ADCP current data located in the Ligurian Sea, Birol et al., 2010 demonstrated that reprocessed altimetry data are able to capture half of the amplitude of the seasonal NC variability. The altimetry currents have then been used to analyze the regional current variability at seasonal scale. In the Balearic Sea, the reliability of altimetry currents has been verified by direct comparison with currents derived

from gliders and HF radars (Bouffard et al., 2010; Pascual et al.; 2015; Troupin et al., 2015). These case studies showed that altimetry can depict current signals coherent with the other instruments. Morrow et al., 2017 also showed that some of the large scale eddies observed by gliders in the

NWMed can be captured by altimetry. A more systematic use of altimetry in regional coastal applications requires a better quantitative assessment of its performance near coastlines, from daily to interannual time scales.

The general objective of this paper is not only to investigate the accuracy of the velocity fields derived from altimetry data next to the coast at different temporal scales, but also to define its contribution compared to the other coastal ocean observing systems which exist in the region (ship-mounted ADCPs, gliders and HF radars). In this study, we combine all the different available in situ data sets which provide information on currents in the Ligurian-Provençal basin and perform systematic comparisons with currents derived from altimetry at different time-scales. In particular, we analyze how the different available observing techniques capture the NC variability and the coherence/divergence/complementarity of the informations derived. From previous studies, we know that only a small part of the NC variations can be captured by conventional satellite altimetry. Here, we use both Jason-2 and SARAL/AltiKa missions to investigate the progress made from Ku-band to Ka-band altimetry. We also investigate the potential of experimental 20/40-Hz altimetry products to monitor the NC variations, relative to the conventional 1-Hz data.

In this paper, section 2 presents the datasets used and the corresponding data processing. It is followed by the intercomparison between the currents derived from altimetry and from the different in situ datasets, with the analysis of the NC variations observed at different time scales by the different instruments (section 3). Section 4 concludes the paper.

## 2. Data and methodology

### 2.1. Satellite Altimetry

We use two altimetry missions with distinct characteristics: Jason 2 and SARAL/AltiKa. Jason 2 was launched in June 2008 and provides long time series of data with a 10-day repeat observation cycle. The performance of SARAL is significantly better. With a better signal-to-noise ratio, it resolves smaller spatial scales than Jason 2: ~40 km against ~50 km (Dufau et al., 2016, Verron et al., 2018). However the corresponding time series started only in February 2013 and have a 35-day repeat observation cycle, a priori not really adapted to the monitoring of the coastal ocean variability. On the other hand, SARAL orbit leads to a smaller distance between tracks, compared to Jason-2 (Figure 1). Here we focus only on the SARAL tracks 302, 343 and 887 and on the Jason 2 track 222, providing the closest data from the in situ observations.

For both missions, because it is one of the most often used in coastal altimetry applications, we used first the X-TRACK regional product from the CTOH (doi: 10.6096/CTOH_X-TRACK_2017_02), processed with a coastal-oriented strategy (Birol et al., 2017). It consists in time series of 1-Hz Sea Level Anomalies (SLA) every 6-7 km along the satellite tracks, available from 20/07/2008 to 01/10/2016 for Jason 2 (i.e. 300 cycles) and from 24/03/2013 to 12/06/2016 for SARAL (i.e. 34 cycles). In order to evaluate the skill of the 20/40-Hz altimetry measurements of the Jason-2/SARAL altimeters for circulation studies, relative to the conventional 1-Hz data, we have also used an experimental high-rate version of these datasets provided by the CTOH (section 3.4). The processing is the same than for 1-Hz SLA, except that the high-rate SLA are computed from the

20/40-Hz range data provided in the AVISO L2 products (https://www.aviso.altimetry.fr/fileadmin/documents/data/tools/hdbk_j2.pdf and https://www.aviso.altimetry.fr/fileadmin/documents/data/tools/SARAL_Altika_products_handbook. pdf). The resulting sea level time series are available every ~0.29 km and ~0.19 km along the satellite tracks for Jason-2 and SARAL, respectively. However, we must keep in mind that if the use of high-rate altimeter measurements allows to significantly improve the spatial resolution, the resulting SLA are much noisier (see for example Birol and Delebecque, 2014). Considering the data availability (see below for the in situ observations), the study period chosen is 2010-2016 for all altimetry datasets.

Jason 2 altimeter is designed as "conventional altimetry" as it operates in the Ku-band frequency. SARAL altimeter operates in the Ka-band, allowing a better performance in terms of spatial resolution (the radar footprint is smaller) and measurement noise. Morrow et al. (2017) analyzed the "mesoscale capability" (defined as the wavelength where the noise is larger than the signal, which varies spatially as shown by Dufau et al., 2016) of these two altimeters in the NWMed using a statistical method (Xu and Fu, 2012). It allows to have an estimate of the size of the structures which can be theoretically detected by each altimeter (in average) and to define the optimal data spatial filtering. Here, we did the same computation for each of the 4 tracks used in this study, using all the data available, unlike in Morrow et al., 2017 where the data located over the continental shelf were discarded. We obtained 49 km for the SARAL track 302, 39 km for the SARAL track 343, 34 km for the SARAL track 887 and 67 km for the Jason 2 track 222, which is coherent with the results of Morrow et al., (2017) who obtained 39 km for SARAL and 55 km for Jason-2 without the coastal altimetry observations. It suggests that the quality of near-shore altimetry SLA remains good. The lower values obtained for SARAL are due to the better signal-over-noise ratio of the AltiKa altimeter compared to Jason-2. The differences obtained between the three SARAL tracks are explained by their respective geographical locations. They capture different mesoscale features.

In order to have the best signal-over-noise ratio, we then filtered the data with a low-pass Loess filter, using a cut-off frequency of 35 km for SARAL. Note that we have chosen a single value for the different SARAL tracks in order to have the same data processing and facilitate the comparison between the different datasets. For Jason-2, we chose the option of using a processing as close as possible from the one of SARAL and then used a cut-off frequency of 40 km. The same low-pass filters were used for both 1-Hz and high-rate SLAs. One need then to keep in mind that noise remains in the filtered Jason 2 data.

Altimetry only provides sea level anomalies relative to a temporal mean. In order to estimate currents as close as possible to the currents measured or derived from the other instruments (see below), we added the regional Mean Dynamic Topography (MDT) from Rio et al., (2014) to the altimetric SLA and computed the surface velocities (u) from the total sea level gradients observed between consecutive points along the track, assuming that the fluid is in geostrophic balance:

$$u = \frac{-g}{f} \frac{\Delta(SLA + MDT)}{\Delta x} \quad \text{, where:} \tag{1}$$

f is the Coriolis parameter, g the gravitational constant and Δx the distance between the points.

Only the across-track component of the geostrophic currents can be derived. The MDT product used is a regional product with an horizontal resolution of 1/16° (i.e. lower than altimetry resolution in the along-track direction). Compared to other MDT products, it allows a better representation of the NC in the Ligurian sea (Rio et al., 2014).

## 2.2. In situ measurements

### 2.2.a) Glider data

Gliders have been deployed in the NWMed since 2005. However, it is only since 2009 that they are regularly operating as part of the MOOSE network (http://www.moose-network.fr/?page_id=272). In particular, on the Nice-Calvi line (Figure 1, pink line), 36 deployments were undertaken between 2009 and 2016. Some of them have already been analyzed in different studies, with different scientific objectives (Piterbarg et al., 2014 focused on the frontal variability, Bosse et al., 2015 investigated the submesoscale anticyclones, Niewiadomska et al., 2008 analyzed physical-biogeochemical coupling mechanisms). Each glider deployment encompassing several transects, the database includes 204 sections; 192 of them are between 2010 and 2016. The ones being too short (<60 km) or moving too far away (>15 km) from an average trajectory computed from the individual ones were discarded. Finally, 173 glider transects along this line were used in this study. It represents a huge amount of observations and a large number of cases available for the comparisons with altimetry or with the other in situ observations.

The campaigns were sliced into ascending (from Calvi to Nice) and descending (from Nice to Calvi) transects and the data were projected on a reference track. We assume that one dive or one ascent represents one vertical profile. In practice, data were discarded when the latitude was not monotonically varying or when the angular deviation between 2 consecutive points and the mean direction of the reference track was too strong (i.e. larger than 3 standard deviations away from the mean angle). Then the data were gridded with a 4 km horizontal bin size along the reference track. 4 km corresponds to the average distance between two successive profiles.

During their mission, gliders measure temperature and salinity from the surface down to 1000 meters (or less if the bottom is shallower, or if commanded to dive shallower). To avoid noise which is mainly due to aliased internal waves, temperature and salinity data have to be filtered. A butterworth filter of second order (Durand et al., 2017) was applied. Different cut-off frequencies have been tested and we finally chose 15 km to avoid noise without removing small-scale variations (as in Bosse et al., 2017). From the temperature and salinity data we computed the density and then the geostrophic velocity component perpendicular to the reference track using the thermal wind equation. These velocities are referenced to 500 m, corresponding to the depth reached by all gliders. The difference with altimetry-derived currents is then that the barotropic component and the baroclinic component below 500 m are missing.

Since 1997, the TETHYS II RV collected a large number of ADCP current measurements during frequent repeated cruises between the French coast (Nice) and the Dyfamed/Boussole site (43°25 N ; 7°52E). The corresponding ship transect is much shorter than the Nice-Calvi glider line (Figure 1), but samples the NC at about the same location. From 1997 to 2014 a 150 kHz ADCP was used, with a vertical bin length of 4 m. In 2015, it was replaced by a 75 kHz ADCP, providing data with a 8 m vertical resolution. The first valid measurement is located at 8/18 m depth for the first/second ADCP. Processed and validated data were obtained from the DT-INSU data center (http://www.dt.insu.cnrs.fr/spip.php?article35). A total of 513 vertical sections of horizontal currents in earth geographical coordinates are provided from November 1997 to March 2017. This number is reduced to 218 during the period 2010-2016. We only used the ADCP transects with a very precise heading which leaves us with 151 sections. Following the same strategy as for glider data, the data were gridded with a 2 km horizontal bin size along a reference transect going from the French coast to the DYFAMED site (green line on Figure 1). Ship tracks located outside the chosen grid bins, incomplete transects, as well as data associated to a ship direction which deviates too much from the reference trajectory (generally corresponding to ship stations) were eliminated. For each cruise, we have one return trip, sometimes two. After a visual inspection of each individual transect to check the coherence of the currents measured during the same day, the data have been averaged per bin, to have one daily-averaged transect. It finally leads to a total of 134 selected current sections. In this study, we focused on the 34 m-depth cell, in order to strongly reduce the surface instrumental errors.

The HF radars data used here (orange zone on Figure 1), are also part of the MOOSE network (Zakardjian and Quentin, 2018). They target the area off the coast of Toulon as a key zone conditioning the behaviour of the NC just upstream of the Gulf of Lions. Due to a sharp bathymetry and several islands that deviate the NC southwestward, significant mesoscale variability and cross-shelf exchanges exist in this area (Guihou et al., 2013), correlated to the strong north-westerlies winds (Mistral, Tramontane). The system consists in two HF (16 MHz) Wellen Radar (WERA) instruments installed near Toulon in monostatic (Cap Sicié station) and bistatic (Cap Bénat/Proquerolles island stations) 8-antenna configurations (see Quentin et al., 2013, 2014 for details). They work with a 50 kHz bandwidth, resulting in a 3 km range resolution, a direction finding method based on MUSIC (multiple signal classification algorithm, see Lipa et al., 2006, Molcard et al., 2009) allowing a 2 degree azimuthal resolution and with a time integration of 20 minutes. The radial velocity maps are averaged over a 1 hour time window and cartesian total velocities are then reconstructed on a regular 2 x 2 km grid. More details on this HF Radar site can be found in Sentchev et al. (2017) who found an overall good agreement between derived radial velocities and in situ ADCP, with relative errors of 1 and 9 % and root mean square (RMS) differences of 0.02 and 0.04 m/s, slightly increased, in velocity and direction, for the reconstructed total velocities, but mainly in conditions of unstationnary wind forcing. The MOOSE HF radar data base used here is made of daily (one diurnal lunar period of 25h), averaged surface currents

computed from the re-processed hourly total velocity data (QC level L3B, i.e., velocity threshold and Geometric Dilution of Precision - GDOP - tests passed) with additional cleaning of residual RFI (Radio Frequency Interference) outliers using outlier-removal algorithm based on the number of L3B valid data, variance and mean over an inertial period window (17 h at 43°N). The data are then filtered from tides and inertial oscillations. The time series starts in May 2012 and ends in September 2014 with a total of 732 days of available data. The size of the area covered by total velocities after the GDOP test is roughly 60x40 km and it is located about 170 km westward of the glider and ADCP observations.

## 2.3 Differences between the currents derived from the different observational techniques

In this study, we extensively compare the currents derived from the four different techniques described above with the objective to better understand how they can optimally complement each other for the observation and study of the variability of the NC circulation system. However, we must first have in mind the intrinsic characteristics of each type of current observation and the differences between the data sets.

- **Spatial and temporal sampling**

First, the locations of the different types of observations do not coincide with each other, and their temporal and spatial sampling is also very different. After processing, current values are obtained every 2 km along the ship ADCP track, every 4 km along the glider line, in a 3 km resolution grid for the HF radar, every 5-6 km / 7-8 km along the satellite track for 1 Hz Jason 2 / SARAL altimetry and every 0.29 km / 0.19 km for HF Jason 2 / SARAL altimetry, respectively. Moreover, each instrument is characterized by specific measurement errors and then specific signal to noise ratio. A filtering has to be applied on the glider and altimetry data, still limiting the wavelengths of the current which can be resolved (see above and in Table 1). We have also to keep in mind instrumental limitations concerning the area which can be monitored. The ship ADCPs, the HF radars and the gliders have a higher spatial resolution than the filtered altimetry data but a much more limited spatial coverage. We have also to consider that the access to altimetry data, at least in the standard 1-Hz version, still remains limited in the 10-15 km coastal band. As the NC fluctuates in both location and width and at both seasonal and much higher frequencies (Albérola et al., 1995), it can make a large difference in the ability of the instrument considered to capture this current flowing along the continental slope, often located very close to the coastline (Figure 2).

Concerning the temporal sampling, the HF radars and the altimetry provide current observations at regular interval: every day for the HF radar product used here, every 10 days for Jason 2 and every 35 days for SARAL. The glider and ADCP data are available between 0 and 9 times per month and between 0 and 5 times per month, respectively. These unevenly spaced time series make the corresponding data analysis more complex since it can produce significant biases in the distribution of the NC properties (as for example its seasonal variations, see Table 2). It will also be influenced by the period of observations available: from about 2 years for the HF radar to more than 6 years for the ADCP, glider and Jason-2 data (see Table 1).

**-   Vertical sampling**

The depth of the current measurement also varies for the different instruments: HF radars and altimeters observe the ocean surface and sub-surface when ADCP and gliders provide vertical sections of measurements. Using both the glider and the ADCP data, we compared the currents computed at different depths (18, 34 and 50 m) and did not find significant differences: less than 5 cm/s for the mean NC core velocity and around 2-3 cm/s for the corresponding STD value. We then decided to use the glider data at 34 m depth to be coherent with the ADCP observations.  We consider that it should not be a significant source of differences with altimetry currents, representing near-surface currents.

**-   Physical content**

Moreover, the different instruments do not capture the same physical content. The ADCP and the HF radars measure both the total instantaneous velocities when the gliders and altimeters allow to derive only the geostrophic current component perpendicular to the satellite or glider track (i.e. excluding the ageostrophic parts such as wind-driven surface current, tidal currents, internal waves, etc…, and the current component parallel to the track). Unlike the other current data sources used here, altimetry gives only access to current anomalies. But the addition of a synthetic MDT allows to overcome this difficulty if its quality is good enough to derive a reliable mean velocity field. After the addition of the MDT, the gliders and altimeters are clearly the closest in terms of current information derived. However, the glider currents are computed from hydrographic measurement profiles with a reference level of 500 m. They miss the barotropic and the deeper baroclinic geostrophic current components when altimetry and MDT allow to estimate absolute geostrophic currents representative of the horizontal density gradients integrated over the whole water column. In this study, in order to minimize the differences between the current data sets, we performed a projection of the ADCP velocities to obtain the current component perpendicular to the ship transects. Concerning the gliders, estimates of depth-average currents computed following Testor et al., 2018 approach were added to the velocity data as an estimation of the barotropic component.

All the differences mentioned above are summarized in Table 1. If the data appear complementary in terms of space-time coverage and resolution, we can anticipate that their respective characteristics make their comparison and combination an issue. It is what will be analyzed in details in section 3.

### 3.   Results

Results below are obtained from 1-Hz standard altimetry measurements, except in section 3.4 which is dedicated to the analysis of the potential of 20/40-Hz altimetry data for coastal circulation studies.

### 3.1.   Mean flow and spatial variability: a regional view

From Figure 1, we can expect that the different observations mentioned above allow to efficiently detect different characteristics of the NC (intensity, position) along its axis, and the variability of these characteristics. In order to have a first general view of how the different velocity fields

compare, we have computed their time-average and their standard deviation values at each point of observation for a common period of time: from March 2013 to October 2014. We need to keep in mind that it corresponds to very different sample sizes: 33 ADCP sections, 8 glider transects, 484 days of HF radar measurements and 54-56 and 16 current data for Jason 2 and SARAL satellite altimetry, respectively. Glider / HF radar observations will then have the lowest / highest significance in terms of statistics. Concerning the HF radars, only the zonal current component is taken into account. Note however that in this area, since the NC is almost zonal, most of its mean and variability are captured in the corresponding statistics. Figure 2 shows the resulting map of the mean current and its standard deviation is in Figure 3. Here, we choose to not represent the results for all the SARAL tracks in order to not overload the figures. Both the regional map (Figure 2a and 3a) and a zoom in the northern Ligurian Sea (Figure 2b and 3b), where the largest number of current observations are located, are shown.

From Figure 1 (see the circulation scheme), we expect negative / positive current values along the northern / southern branch of the cyclonic NC system. It corresponds to what is observed in Figure 2, where one can notice a very good consistency of the mean currents derived from all the different instruments. Putting together all the pieces of information, the regional structure of the circulation emerges. As already shown in Birol et al. (2010), in the Tyrrhenian Sea, the northwestward Tyrrhenian Current (TC) is well observed at the northern end of Jason 2 track 161. Further north, the NC is formed by the merging of the Eastern Corsica Current (ECC), captured just east of Corsica by the Jason 2 track 085, and of the Western Corsica Current (WCC), well captured by both the gliders and the SARAL track 343. The WCC appears however more extended towards the open sea in the SARAL data, compared to the glider. The NC is then strongly constrained by the bathymetry and follows the continental slope along the coasts of Italy, France and Spain. It can be continuously followed from the SARAL track 343 to the Jason 2 track 070, through the ADCP, glider and HF radar observations. Mean NC velocities larger than -0.3 m/s are observed in the Ligurian Sea by ADCPs and Jason 2 altimetry, and off Toulon by the HF radars. Then the continental slope current slows down offshore the Gulf of Lion: the Jason 2 track 146 gives a mean current value of ~-0.15 m/s. Its flow is then almost divided by three in the Balearic Sea (~-0.10 m/s). Further south, around 40.5°N around 5-6°E and then between 42°N and 42.5°N around 7-8°E, an eastward flow, probably associated to the Balearic Front which closes the cyclonic circulation south of the Northwestern Mediterranean basin, is captured by the Jason 2 tracks 146, 009, 222 and the SARAL tracks 302 and 887, from west to east. Around 8°E, it slightly deviates to the southeast before joining the WCC.

If we focus on the northern Ligurian Sea (Figure 2b), the cross-track direction of Jason 2 track 009 is not well oriented compared to the local axis of the NC. In this area, the continental shelf is very narrow and as a consequence the NC is very close to the coast: altimetry struggles to observe the corresponding flow. However, the Jason 2 track 009 and SARAL track 887 still capture a westward current at their northern end. Considering altimetry, Jason 2 track 222, located further southwestward, appears better oriented to monitor the NC. In this area, despite the difference in the number of data samples, the altimetry, ADCP and glider mean current values are very close: between -0.24 and -0.32 m/s for all of them. The width of the NC tends to vary from one instrument to the other. With the gliders it appears slightly narrower than with the ADCP and altimetry (i.e.

SARAL track 887). Note also that the ADCPs and gliders, which provide more nearshore information, show a positive or almost null flow very close to the coast, not observed by altimetry which stops further offshore. Still further west, the altimetry and HF radars also capture a coherent mean NC flow, but with larger values in HF radars (~-0.44 m/s) than in altimetry (~-0.28 m/s). This difference is probably due to the ageostrophic motions captured by the HF radars but not by altimetry, and to the differences in the data resolution.

Figure 3 represents the associated current variability, as captured by the different types of observations. Not surprisingly, in all datasets, larger standard deviation values generally coincide with the NC system. In altimetry, we observe values of 0.12-0.2 m/s at the northern ends of the Jason 2 tracks 161, 085, 044, 222, 146 and 070 (the signal at the end of track 146 does not correspond to the NC) and of SARAL tracks 302, 343 and 887. If we focus on Figure 3b, on Jason 2 track 222, we first see clearly the coastal current variations associated to the NC flow (see also Figure 2b). However, the NC is not fully resolved by 1 Hz-altimetry data: observations stop at ~10 km from land. The more coastal observations have been discarded during the processing, probably due to large data errors. This is even more true for the Jason 2 track 009 (the last data point available is associated to a large suspicious current value) and the SARAL tracks 887 and 302. We have to keep in mind that in this area, where the narrow NC flow is very close to the coastline (its core is in the range 10-40 km from land, Piterbarg et al., 2014), its observation by altimetry is very challenging. In comparison, the ADCP, glider and HF radar data allow to observe the NC variability much closer to the coast: our datasets stop at 2.5 km, 3.5 km and 3-7 km from land, respectively. But they all differ in the current variance captured. Concerning the ADCPs and gliders, observing the NC at the same location, the ADCPs show larger standard deviation values (~0.13 m/s) almost all along the transect when the gliders show much lower values in the open ocean (~0.05 m/s), increasing on the shelf break to values very close to the ones observed on Jason 2 track 222 (~0.15-0.20 m/s). Further west, the HF radars show the largest current variance south of Toulon, with values around 0.23 m/s located on the continental shelf break. In comparison, the corresponding NC variance captured by the SARAL track 302 is only half of that. Further south, off Corsica, the gliders show very low variability, roughly half of the values corresponding to the NC, indicating a WCC flow which is very stable in time (as also shown in Astraldi and Gasparini, 1992).

Considering the intrinsic and important differences between the different current datasets (section 2.2.d), these first statistical results are encouraging. They give a coherent picture of the regional circulation, with, except for the HF radars which capture a faster current flow, about the same NC average velocity values. The NC variability is also clearly captured by the different data sets all along its path, but with significant differences in terms of amplitude. Note that when we recompute the standard deviations using a larger period of time (not shown), ADCP and glider tend to converge toward the same cross-shore profile than the one derived from Jason 2 track 222, with a maximum which is about 0.03 m/s larger for the in situ observations. We can then conclude that this diagnostic is largely influenced by the number of data samples considered as well as by the period of time covered by the measurements.

In order to better understand the differences in variability captured by the various data sets, we analyze the time-space diagrams of the currents derived from ADCP, HF radar, glider and altimetry data over the period considered (Figure 4). We focus on the first 60 km off the French coast and, concerning altimetry, on SARAL tracks 302 and 887 and on Jason 2 track 222. The HF radar data correspond to a meridional section of the zonal current component located at 6.2°E. The NC is clearly detected in all data but Figure 4 displays large variations at different timescales (see also Font et al. 1995; Sammari et al., 1995; Albérola et al., 1995) that make the data temporal sampling resolution a very sensitive question if we want to study this current system. The number of glider transects is low and concentrated in 2013 and the unevenly spaced ADCP sections miss a large number of events. Spring 2013, winter and summer 2014 are poorly sampled. The HF radar provides a very good temporal sampling according to the one needed to capture the high-frequency NC variations but it monitors only its section located in the vicinity of Toulon. Altimetry provides then a good complementary information. Despite its relatively low spatial resolution and the intrinsic difficulties when approaching the land, it detects seasonal changes coherent with the ones observed in the other data sets as well as much shorter period changes. Note that if the SARAL mission capabilities are expected to be particularly adapted for fine-scale oceanography and coastal applications (Verron et al., 2018), in our case study its 35-day period appears to be a strong limitation to monitor the highly fluctuating NC flow. This particular point will be further analysed in section 3.3. In the next section, we concentrate on the seasonal variability observed in the different data sets, as it is known to be the dominant signal of the NC system at regional scale (Alberola et al., 1995; Sammari et al., 1995; Crépon et al., 1982; Birol et al., 2010).

## 3.2. The seasonal variability of the NC flow captured by the different instruments

Here we compare the monthly climatology (i.e. the mean value for each month of the year) of the maximum NC amplitude computed from the different current data sets (ADCP, glider, HF radar and altimetry). This time, we use all the data available during the period 01/01/2010 - 31/12/2016 (note that the HF radar data are only available over the period 2012-2014). Concerning altimetry, we consider only Jason 2 since we have 2-4 samples per month for SARAL, which is not enough to compute meaningful statistics (see Table 2). For each data sample available, the current profiles along the Jason 2 track 222, the ADCP and glider reference transects and a meridional HF radar section located at 6.2°E, are analyzed. The maximum NC amplitude is defined as the average of the first decile of the velocity values for each transect and time (remember that the NC corresponds to negative current values). These values must be close in space. This strategy allows to filter large isolated current values which may not correspond to the NC. In altimetry, only a distance spanning 60 km to the coast is considered. The number of data in the first decile varies according to the data set and to the number of data in the section considered. Because of the lower resolution, it always corresponds to one point in altimetry. As we can see in Figure 4d, data gaps exist in Jason 2 for some cycles. When more than 3 points are missing, the corresponding cycle is discarded from the analysis. Finally, all the maximum NC values collected are averaged as a function of month and data set and synthesized in monthly climatologies. The results derived from in situ data are in

Figure 5a and the results derived from altimetry are in Figure 5b. The glider results are on both figures because this instrument provides the currents which are the closest to altimetry in terms of physical content. For each month, the standard deviation computed from all the NC amplitude values available is also indicated.

Table 2 lists the temporal distribution of the number of samples included in the calculation as a function of month (in brackets). The data density is much more important than in section 3.1 and the corresponding statistics more robust. It appears relatively stable for Jason 2 altimetry and more heterogeneous for the other observations. The number of in situ data per month is strongly variable, especially for the ADCP and to a lesser extent for the glider, and varies also a lot from one year to
the other. 24 ADCP transects are available in 2015 and only 7 in 2012 and 2014, when the glider dataset has a large gap in 2014. As a consequence, the results will be only discussed in terms of seasonal tendencies.

In Figure 5a and b, except altimetry, all the climatologies show a clear and coherent seasonal cycle
of the NC amplitude, with a stronger / lower flow in winter / summer. As already seen in the previous section, compared to the other data sets, the HF radars capture a faster NC south of Toulon. Higher NC velocities are expected in this location (Ourmières et al., 2011). The corresponding amplitude of the seasonal variations is 0.32 m/s, with a minimum of -0.34 m/s in August and a maximum of -0.66 m/s in February. These values are also found by Guihou et al.,
2013 in the same area. In comparison, further East in the northern Ligurian Sea, the peak-to-peak amplitude of the seasonal cycle is slightly lower for the ADCPs than for the HF radars, and associated to a lower mean flow, with a minimum of ~-0.27 m/s in August and a maximum of -0.54 m/s in January. Note however that the value observed in January may be less robust (or at least poorly representative of a mean monthly situation) since it is computed only with 3 data samples.
Concerning the gliders, the peak-to-peak amplitude variation is ~25% lower than for the ADCPs, with a minimum of ~-0.25 m/s in August/September and a maximum of -0.46 m/s in December. Since these instruments measure velocities at very close locations, the differences may be mainly due to ageostrophic currents. The Jason 2 climatology displays significantly different results with a series of maxima (~-0.46 m/s in February and November) and minima (~-0.35 m/s in May and
October).

For further analysis, we consider the dispersion of individual current values for each month (Figure 5a, b, envelopes around the curves). We observe significantly different date-to-date variability for each month: between 0.03 and 0.15 m/s for the glider and ADCP, between 0.12 m/s and 0.20 m/s for the HF radar and between 0.08 and 0.17 m/s for altimetry. It indicates that the seasonal NC cycle
observed in Figure 5 is modulated by a strong mesoscale and/or year-to-year variability, and it seems to be especially true during intermediate seasons. The dispersion curve of Jason 2 generally follows the other ones except in July and September, when it shows large peaks of variability. Deeper inspection in the corresponding current data set reveals that it is due to much larger NC amplitudes observed during these months in 2014 and 2015. The corresponding NC intensifications
are clearly observed in Figure 4d in July and September 2014. Unfortunately, no glider transect is available during these periods (Figure 4c) and we have only one ADCP section which does not

show a NC flow increase (Figure 4b). However, the HF radar currents (Figure 4f) tend to support that the NC intensification captured by Jason 2 is realistic and not due to altimetry errors. One profile of SARAL track 887 is available in July 2014 and it observes the same feature (Figure 4a). Since we did not find evidence of summer NC intensification in the previous years, we decided to recompute the seasonal cycle of the NC amplitude using only the data available during the first 6 year-period of Jason-2 (i.e. 2008-2014). We did the same for the ADCPs and gliders, but very few glider data and no ADCP currents are available before 2010. HF radar currents have not been considered because of the too short length of the time series. The resulting curves are shown in Figure 5c and a clear seasonal cycle is now also observed in the climatology derived from Jason 2, with a summer / winter decrease / increase of the NC flow. Note that it is also coherent with the results of Birol et al. (2010) who used a combination of the T/P and Jason-1 altimeter missions to obtain a current time series over the 1993-2007 time period. The amplitudes of the seasonal variations computed during this new period of time are now around 0.29 m/s, 0.27 m/s and 0.16 m/s for the ADCP, glider and Jason2 altimetry data, respectively. Figure 5c highlights that the summer velocities measured by the in-situ instruments are relatively close on average. During winter and especially spring, the differences become significant in both amplitude and phase.

Two physical processes can explain that the differences between the different types of current measurements vary as a function of the season. First, the stronger mesoscale variability associated to the NC during winter and spring makes the space and time sampling of the current measurements a more critical issue for the study of this current system at that particular time of the year. Second, the strong Tramontane and Mistral winds are more frequent in winter and spring. Then, the differences between the glider and the ADCP current measurements, very close in location, may be more important when the non geostrophic dynamics (in particular the Ekman flow) produced by the strong winds is more important. The closest seasonal variations to the ones observed by altimetry are found for the glider. It is not surprising since the currents derived from this instrument are also the closest in terms of physical content (see section 2.2.d). Despite the spatial resolution of altimetry data and the width and very coastal location of the NC current, the amplitude of its seasonal variations captured by the Jason2 track 222 along the French coast is 55-60% of the amplitude captured by both the gliders and ADCPs.

### 3.3.  Individual snapshots

To learn more about the similarities and differences between the currents derived from the different instruments, as well as their causes, we now analyze the observations at particular dates. In order to minimize, as far as possible, the differences due to distances in space and time between observations, we focus here on the region near Nice (i.e. on the ADCP and glider data, as well as on the SARAL track 887 and the Jason 2 track 222), and consider only observations that are close in time. For each day of the 2010-2016 study period, we used a time window for each data set: 5 days for Jason 2, 10 days for the glider and ADCP data and 22 days for SARAL. We selected only the dates for which we had the four types of observations available and finally obtained 7 cases which are reported in Table 3. The corresponding cross-track currents are shown in Figure 6 (by season) as a function of the distance to the point where the corresponding transect intersects the coastline. For each case and each data set, we have computed the maximum NC amplitude following the same

method than in section 3.2, and the corresponding location. The latest is expressed in terms of distance to the coast. The results are provided in Table 4.

Figure 6 highlights very different NC situations. Here, the largest coastal current velocities are observed in spring and not in winter as expected from section 3.2. Case 1 (Figure 6a), the only one in this season, shows by far the strongest NC amplitudes in ADCP and glider data (< -0.6 m/s), associated to a narrow flow located within the 30 km coastal band. It corresponds to a difficult study case for altimetry which is still able to depict the NC, but with a too large current vein which amplitude is less than half of what is observed in the in situ observations. Cases 2 and 4 (Figures 6b, c) are in summer. The NC is broader and its velocity is around -0.3 m/s in all data sets, except in the glider of case 4 (see below). This time, altimetry successfully captures the NC amplitude; the location of its core is also good in case 4 but not in case 2 (it is too far to the coast for SARAL). In case 4, altimetry and ADCP currents are very close but, for a reason which is unclear (it may be due to a NC meander or eddy captured by the glider and not by the other instruments), the glider represents a significant slower flow located further south. Cases 5 and 6 (Figure 6d, e) correspond both to autumn situations but they highlight very different coastal current patterns. In case 5, the glider and SARAL data, corresponding to the same day, are very coherent: they show a relatively weak NC flow (~-0.2 m/s) which core is ~30 km to the coast. Jason 2 observations, very close in time to SARAL and the glider data, show a larger current located slightly further south (~6 km). The ADCP represents a NC vein at the same location than in the glider and SARAL but with a much stronger amplitude. It could be due either to the differences in the dates of observations (one week from Table 3, temporal scale at which meanders develop) or to an important ageostrophic NC component. In case 6, a lack of data for Jason 2 can be observed which lead to question the realism of the current estimates close to the coast. However, the glider, the ADCP and SARAL data show a broad NC located further offshore than in the other cases. Its core is located ~ 40 km offshore in ADCP and glider data. As in case 5, the glider and SARAL data provide NC amplitudes and locations that are relatively close and the ADCP data give a larger NC maximum. A particular feature in this autumn situation is the succession of very strong and narrow southwestward and then northeastward flows observed in the first 20 km coastal band in both ADCP and glider currents. It is not captured by SARAL which does not get close enough to the coast. It is probably associated to an eddy or meander sticked on the northern anticyclonic side of the NC (eddies were documented at this location in Casella et al., 2011). Finally, cases 3 and 7 (Figures 6g, f) correspond to winter situations and, as for the autumn, they are very different. In case 3, we observe a broad NC with a core located around 30 km to the coast. The glider exhibit current oscillations along its transect but all current data sets show a coherent representation of the NC, even if the ADCP data provide larger velocities. In case 7, the glider and ADCP capture a narrow NC located ~20 km off the coast also observed by altimetry but with some differences: in Jason 2 the NC flow is not entirely captured and in SARAL it is located further offshore. It may be due to rapid variations of the NC between the different dates of observations: 12 days between ADCP and SARAL.

Beyond the large variations of the NC characteristics from one case to the other, an interesting feature in Figure 6 is the presence of an eastward flow located south of the NC (i.e. 100-150 km to

the coast) in altimetry data in different cases (cases 4, 5 and 6 in particular). The ADCP transect is too short to capture this current vein and it is not observed in the glider data, located further east compared to SARAL track 887 and Jason 2 track 222. The latter rather depict the WCC on the southern edge of its section. To our knowledge, this offshore eastward flow is not documented in the literature but its signature seems also be observed in Figure 2a and 3a (around 42.5°N in SARAL, and around 42.8°N in Jason2). It will be further discussed in the section 3.5.

Finally, what is illustrated in Figure 6 is that, because of the large short-term changes in the NC circulation system, each snapshot of observations differs significantly from the corresponding seasonal average. It highlights the strong interest of long-term and regular altimetry data to study the persistent components of the NC circulation system, as well as its seasonal variations and possible longer-term changes.

### 3.4 Can we improve the estimation of the NC characteristics with high-rate altimetry compared to 1-Hz data?

In this section we consider the improvement that is possible to obtain in terms of current derivation with the use of high-rate altimetry measurements, compared to the conventional 1-Hz data used above. However, if research coastal altimetry products, calibrated and validated and covering different regions and missions, are now available at 1-Hz, it is not the case for high-rate altimetry products. Even if some studies have shown the better performance of 20/40-Hz altimeter measurements to observe the coastal circulation (Birol and Delebecque, 2014; Gomez-Enri et al., 2016) they are much noisier and no consensus exists yet concerning their (post-)processing. Here, we used an experimental version of high-rate X-TRACK SLA data for both Jason-2 and SARAL which original measurements are at 20-Hz and 40-Hz, respectively. Since a lot of erroneous data remained in the coastal area, we applied a 2-sigma filter on the resulting SLA fields along each individual track and cycle, in order to edit the data before filtering and then computation of the current estimates (section 2.1).

In order to analyze if we can expect a better observation and understanding of the NC variations from high-rate altimetry measurements, they have been used to compute the same diagnostics than in sections 3.1, 3.2 and 3.3. Only the results for the individual snapshots will be illustrated here (Figure 7) since, even if the major difference with the current fields derived from 1-Hz altimetry is that the larger number of coastal data allows to estimate currents closer to the coast and then to better resolve the NC flow (see Figure 7), we did not find significant differences in the NC statistics (i.e. mean current and standard deviation values) and amplitude of the seasonal cycle computed from 20/40-Hz SLA, compared to the 1-Hz solutions.

In Figure 7, the same color code than in Figure 6 is used. For each case, as in section 3.3, the maximum NC amplitude and corresponding location are reported in Table 4 for both SARAL and Jason-2. In case 1 (Figure 7a), the gain obtained with the use of HF data is very clear. A this date the NC vein is narrow and located near the coast. Contrary to the 1-Hz solution, the NC is better resolved by both SARAL and Jason-2 high-rate altimetry. It is especially true for SARAL with NC characteristics that are almost identical to the ones derived from the gliders. In Jason-2, the NC core

is also close to the glider solution but its amplitude is ~35% lower. For cases 2 and 4 which correspond to the summer (Figure 7b and c), here again the use of high-rate altimetry allows a better observation of a NC vein but the agreement with in situ data is not so good. Concerning SARAL, for case 2, the current estimates are suspect since a reduction of the current intensity appears at the location of the NC core in the other datasets (Figure 7b). For case 4, SARAL NC amplitude is too high: 0.55 m/s vs 0.16 m/s for the glider. Jason-2 high-rate NC estimates appear closer to the in situ data than SARAL for both cases 2 and 4 but the resulting NC characteristics do not appear better than the corresponding 1-Hz estimations, when compared to ADCP (Table 4). It probably reveals that the cut-off frequency chosen in the filtering is too low. Cases 5 and 6 (Figures 7 d and e) also show some very doubtful oscillations in both SARAL and Jason-2 currents and high-rate altimetry does not improve the NC estimations. In winter, the cases 3 and 7 (Figures 7g and f, respectively) are very different. In case 7, 20-Hz Jason 2 data depicts the entire NC with current estimates much closer to in situ data (especially the glider), compared to 1-Hz Jason-2 measurements. In case 7 they degrade / improve the NC representation (Figure 7g) if we refer to the glider / ADCP, respectively. Note that this case illustrates the difficulty of the calibration of altimetry data processing algorithms with independent observations since results may differ as a function of the independent observations used. Here, 40-Hz SARAL data show a too noisy current solution.

As already shown in previous work (Birol and Delebecque, 2014; Gomez-Enri et al., 2016), high-rate altimetry allows to derive significantly more sea level data near the coast. Here we observe that the coastal circulation derived is better resolved in space, both in terms of horizontal resolution and of distance to the coast of the current estimates. However, the resulting current fields depend crucially on the strategy followed for data processing, -including retracking and corrections-, screening and filtering.

## 3.5 The seasonal variability of the regional surface circulation observed by altimetry

Here we use only 1-Hz altimetry data. In order to separate the seasonal component of the surface circulation from the mesoscale variations, along each pass of Jason 2 and SARAL located in the area of interest, we have computed a seasonal "climatology" of the cross-track surface geostrophic currents captured by these two altimetry missions (Figure 8). It was done by simply averaging the corresponding seasonal velocity values for the common 3-year period: April 2013 – April 2016. Note that this type of analysis can be already found in Birol et al. (2010) with a much longer period of altimetry data, but with Jason measurements only. The need to use multi-mission observations was incidentally pointed out in this study. Here, indeed, the combination with SARAL data largely improves the spatial resolution of the regional circulation, enabling to capture the main current veins at much more locations along their path (see Figure 9 of Birol et al., 2010 for comparison). In Figure 8, all the structures of the standard circulation scheme of the NW Mediterranean Sea (Figure 1) are observed: the NC, the WCC, the Balearic Current, the Balearic Front and the TC. What can also be noticed first is the very good coherence and complementarity between the SARAL and Jason 2 climatologies, especially at crossover points. The seasonal variations of the regional circulation system already discussed in details in Birol et al. (2010) are confirmed from this

different and shorter period of altimetry observations. In particular, if a stronger and unique southwestward flow is observed along the Italian, French and Spanish coasts from autumn to spring, it is not so clear during summer. During this season, the NC does not seem to continue west of 4°E to reach the Balearic Sea. Instead, it may recirculate eastward offshore Cape Creus.

More generally, compared to Birol et al. (2010), the better spatial coverage obtained by combining both SARAL and Jason 2 reveals a circulation scheme that could be much more complex than the one classically proposed in the literature. In summer and autumn (Figure 8a,d), between 3°E and 9°E, individual eastward current veins are observed between the NC and the Balearic Front, suggesting that recirculations may exist along its path during these seasons. One of them corresponds to the eastward current branch mentioned in section 3.3. Note however that this seasonal analysis is based only on 3 years of observations and could be biased by particular features occurring during 2015. Further investigation based on numerical modeling is clearly needed. This is the next step of this study. Here again, altimetry appears clearly as a very good tool to first validate the model results.

## 4.   Discussion and conclusion

The characteristics of the dynamics as well as the diverse arrays of in situ instrumentation in the NWMed offers the possibility to evaluate in details the complementarity between different types of measurements to monitor the coastal ocean circulation. In this study, the systematic comparison of the current data derived from the different platforms provide new insights into the biases that their differences cause in the estimations of the NC characteristics. Compared to previous studies comparing altimetry and in situ observations, the originality of this study comes from the number of instruments and observations used, as well as from the long time period addressed and the area covered. It demonstrates that altimetry can be integrated in multi-platform coastal current monitoring systems and enables to analyze the relative capability of each type of instrument.

The HF radars provide a good daily view of the NC but only for a small area (60x40 km) and, as they observe only the surface layer, the NC can be hidden by a strong Ekman flow. The ship-mounted ADCP allows to see the vertical NC structure at very high resolution and up to the coast but it is irregularly sampled and the measurements may contain unsteady ageostrophic current components such as inertial oscillations (Petrenko et al., 2008). Since they can be operated on a routine basis only in a few number of places, we have only one regular section crossing the NC off the French coast and it is relatively short. It is also the case of gliders which horizontal resolution and temporal sampling is lower than that of the ADCP and the HF radars but which provide much longer sections of observations. More generally, they also allow to measure a large number of physical and biological ocean parameters. Alongtrack altimetry provides a reasonably good monitoring of surface currents in both space and time but its effective spatial resolution (section 2.1) does not allow to resolve all the mesoscale and sub-mesoscale signals associated to the NC. Here, the combination of all the observations derived from the different instruments highlights the continuity of the NC from the Italian coast up to the Spanish coast. The general coherency between

the different current estimations enables also to go one step forward in the quantification of the NC components that can be observed in altimetry.

If we consider a reasonably long time series of observations including enough data samples for each instrument (see section 3.2), in the northern Ligurian Sea, the average NC value derived from altimetry is -0.3m/s and is coherent with the estimations derived from the other instruments. Concerning the amplitude of its seasonal variations, it is underestimated by ~40-45% in average, compared to both the glider (closest instrument in terms of physical content of current estimations) and the ADCP (the highest resolution current data set). Altimetry-derived currents will miss a larger part of the absolute surface velocity field in winter and spring, compared to summer and autumn, because the Ekman component and finer-scale motions are more important during that seasons. For individual dates, the NC component that is not observed by altimetry varies a lot from a correct NC amplitude estimation to no NC observation, as a function of the location (i.e. more or less close to the coast) and width of the NC.

This study enables also to compare the relative performance of two generations of altimetry missions and of both 1-Hz and high-rate measurements. It confirms that the standard 1-Hz along track altimetry products derived from Ku-band radars provide meaningful estimations of the NC (as already shown in Birol et al., 2010 and Birol and Delebecque, 2014). The new Ka-band SARAL altimeter data tend to give estimations of the NC characteristics that are closer to in situ data in a number of cases but its 35-day cycle is clearly a strong limitation for the study of this coastal current system. The use of 20-40Hz altimetry measurements improves significantly the number of near-coastal sea level data and then the resolution of the NC. However, the currents derived are still relatively noisy, meaning that their (post-)processing strategy is still at an experimental stage and needs to be improved.

Not surprisingly, another conclusion of this study is that the data resolution and sampling is clearly an issue to capture the large range of frequencies found in the NWMed coastal ocean (and we can easily assume that it is true for many other coastal ocean areas). In particular, the temporal data coverage is a large source of differences between the NC statistics computed from the different observing systems. A second cause of differences in the estimations of the NC characteristics appears to be due to ageostrophic flow, principally the Ekman and inertial currents, measured by the ADCP and HF radars but not represented in the glider (even if they can be partially included through the correction of the depth-averaged currents) and altimeter-derived geostrophic currents. Clearly, a multi-data combined approach is the unique way to obtain a complete picture of a dynamical system as complex as the NC.

Finally, it is important to note that improved altimetry data processing and corrections as well as technical innovations lead to an ever increasing number of coastal data ever closer to the coastline. It raises the question of the calibration and validation of these new data against independent in situ observations. How can we robustly quantify the evolution of the new processing and products? We benefit from the long experience of nadir altimetry technology, widely based on tide gauges sea level observations taken as an independent reference. However a full understanding and exploitation of the new performances allowed by the Ka-band, SAR and SAR-in altimetry techniques, as well as by the use of high-rate altimetry measurements, requires new methods and validation means. We advocate that only a combination of in situ instruments providing regular

cross-shore informations along altimetry tracks will allow to understand and exploit the full capability of altimetry in coastal observing systems and guide its evolution.

*Acknowledgements*.

This study was done with the financial support of the Region Occitanie and the CNES through their PhD funding programs. The ship-mounted ADCP data were kindly provided by the DT-INSU who did the acquisition, management and processing. Altimetry data used in this study were developed, validated, and distributed by the CTOH/LEGOS, France. Glider data were collected and made freely available by the Coriolis project (http://www.coriolis.eu.org) and programs that contribute. We would like to acknowledge the staff of the French National Pool of Gliders (DT-INSU/CNRS - CETSM/Ifremer) for the sustained gliders deployments carried out in the framework of MOOSE. Support was provided by the French ''Chantier Méditerranée'' MISTRALS program (HyMeX and MERMeX components) and by the EU projects FP7 GROOM (grant agreement 284321), FP7 PERSEUS (grant agreement 287600), FP7 JERICO (grant agreement 262584), and the COST Action ES0904 ''EGO'' (Everyone's Gliding Observatories). HF radar data were funded as part of the French MOOSE Mediterranean observing system program. Finally, we thank the anonymous reviewers for their constructive comments, which helped to improve the quality of the manuscript.

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

**Table 1: Main characteristics of the different current data sets used in this study.**

| Instrument | Physical content | Depth | Spatial resolution | Temporal resolution | First and last dates in the data record | Number of sections selected | Filtering |
|---|---|---|---|---|---|---|---|
| HF radars | Absolute surface current | surface | 3 km | daily | May 2012 - September 2014 | 732 | No |
| ADCP | Vertical section of absolute current | 34 m chosen for this study | 1.3 km | unevenly spaced : 1 day to 6 months between consecutive data | May 2010 - November 2016 | 134 | No |
| Gliders | Vertical section of geostrophic current (baroclinic component above 1000 m + additional correction) | 34 m chosen for this study | 4 km | unevenly spaced : 1 day to 1 year between consecutive data | June 2010 - September 2016 | 173 | 15 km |
| Jason 2 1Hz (20 Hz) | Surface geostrophic current | near-surface | 5.75 km (0.29 km) | ~ 10 days | January 2010 - October 2016 | 246 | 40 km |
| SARAL 1Hz (40 Hz) | Surface geostrophic current | near-surface | 7.38 km (0.19 km) | 35 days | April 2013 - May 2016 | 34 | 35 km |

**Table 2: Number of data sample per month for each current dataset during the period 01/01/2010 - 31/12/2016. The number of data selected for the climatology computation is indicated in brackets.**

| Instrument | Jan | Feb | Mar | Apr | May | June | July | Aug | Sep | Oct | Nov | Dec |
|---|---|---|---|---|---|---|---|---|---|---|---|---|
| radars | 62 (60) | 56 (55) | 62 (62) | 60 (60) | 93 (70) | 90 (90) | 93 (91) | 93 (52) | 90 (70) | 62 (35) | 60 (29) | 62 (53) |
| ADCP | 6 (3) | 20 (11) | 18 (5) | 20 (10) | 15 (8) | 25 (9) | 18 (11) | 24 (15) | 20 (12) | 11 (6) | 24 (15) | 17 (8) |
| Gliders | 6 (6) | 20 (20) | 12 (10) | 12 (12) | 10 (10) | 28 (23) | 26 (22) | 14 (14) | 10 (9) | 17 (15) | 17 (14) | 20 (16) |
| Jason 2 | 22 (22) | 20 (20) | 21 (21) | 20 (20) | 21 (21) | 21 (20) | 22 (22) | 22 (22) | 20 (19) | 19 (19) | 18 (18) | 19 (19) |
| Saral | 3 | 3 | 2 | 3 | 4 | 4 | 2 | 2 | 3 | 3 | 3 | 2 |

**Table 3: List of the cases of relative colocalisation in time between the glider, ADCP and atimetry current data, and corresponding dates of observations.**

| | Date of observations | | | | |
|---|---|---|---|---|---|
| | Glider | ADCP | SARAL altimetry (track 887) | Jason 2 altimetry (track 222) | Temporal window |
| Case 1 (Figures 6a and 7a): April 2013 | 11-13/04/2013 | 11/04/2013 | 14/04/2013 | 11/04/2013 | 4 days |
| Case 2 (Figures 6b and 7b): July 2013 | 12-14/07/2013 | 13/07/2013 | 28/07/2013 | 09/07/2013 | 20 days |
| Case 3 (Figures 6g and 7g): February 2015 | 6-15/02/2015 | 09/02/2015 | 08/02/2015 | 04/02/2015 | 12 days |
| Case 4 (Figures 6c and 7c): September 2015 | 18-26/09/2015 | 22/09/2015 | 06/09/2015 | 20/09/2015 | 21 days |
| Case 5 (Figures 6d and 7d): October 2015 | 06-11/10/2015 | 17/10/2015 | 11/10/2015 | 10/10/2015 | 12 days |
| Case 6 (Figures 6e and 7e): November 2015 | 13-21/11/2015 | 12/11/2015 | 15/11/2015 | 18/11/2015 | 10 days |
| Case 7 (Figures 6f and 7f): February 2016 | 1-9/02/2016 | 05/02/2016 | 24/01/2016 | 27/01/2016 | 17 days |

**Table 4: Maximum NC value and distance to the coast of this maximum deduced from the glider, ADCP and atimetry current data for the 7 individual cases listed in Table 3.**

| | Maximum NC value and distance to the coast of this maximum | | | | | |
| --- | --- | --- | --- | --- | --- | --- |
| | Glider | ADCP | Saral altimetry (track 887) | | Jason 2 altimetry (track 222) | |
| | | | 1 Hz | 40 Hz | 1 Hz | 20 Hz |
| Case 1 | 18 km -0.66 m/s | 23 km -0.74 m/s | 21 km -0.29m/s | 19 km -0.66 m/s | 19 km -0.34 m/s | 22 km -0.42 m/s |
| Case 2 | 22 km -0.27 m/s | 28 km -0.31 m/s | 50 km -0.30 m/s | 37 km -0.20 m/s | 24 km -0.39 m/s | 31 km -0.41 m/s |
| Case 3 | 30 km -0.30 m/s | 33 km -0.51 m/s | 29 km -0.26 m/s | 50 km -0.34 m/s | 24 km -0.34 m/s | 24 km -0.41 m/s |
| Case 4 | 42km -0.16 m/s | 18 km -0.28 m/s | 21 km -0.31 m/s | 17 km -0.55 m/s | 19 km -0.37 m/s | 23 km -0.35 m/s |
| Case 5 | 23 km -0.22 m/s | 26 km -0.42 m/s | 21 km -0.23 m/s | 43 km -0.33 m/s | 30 km -0.53 m/s | 31 km -0.49 m/s |
| Case 6 | 30 km -0.25 m/s | 39 km -0.40 m/s | 29 km -0.26 m/s | 25 km -0.34 m/s | 19 km -0.26 m/s | 23 km -0.62 m/s |
| Case 7 | 14 km -0.30 m/s | 16 km -0.42 m/s | 43 km -0.44 m/s | 46 km -0.28 m/s | 10 km -0.54 m/s | 15 km -0.37 m/s |

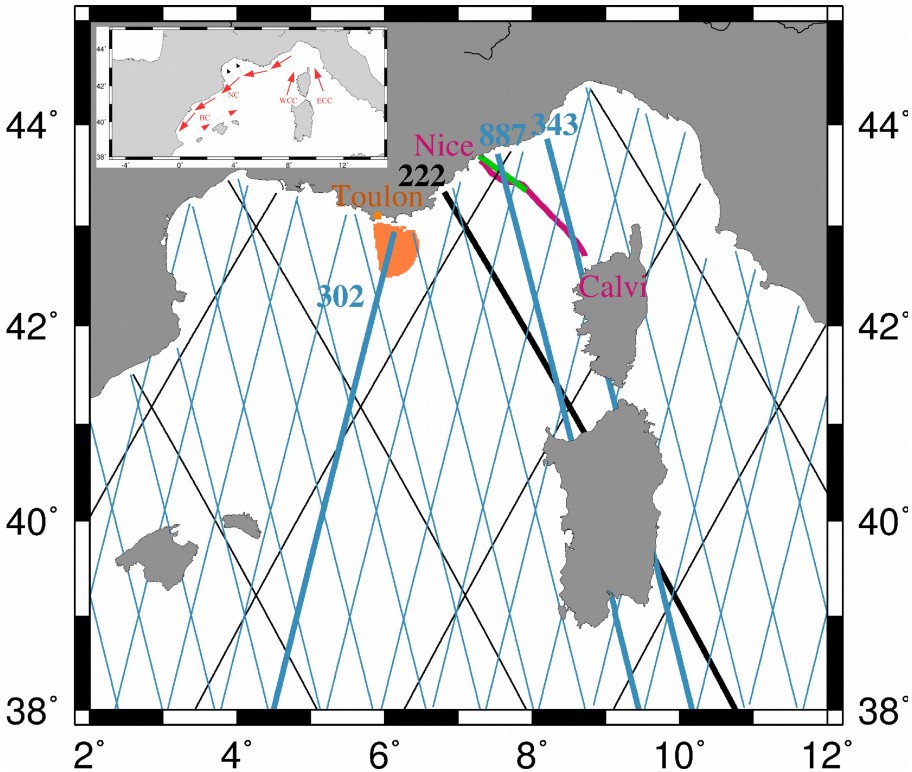

**Figure 1: Study area and data distribution. Jason 2 and SARAL tracks are represented by the black and blue lines, respectively. The satellite tracks used in the study are indicated in bold. The region in orange corresponds to the HF radar coverage. The Nice-Calvi glider line is in purple and the Thetys ADCP transect is in green. A map of the schematic regional circulation is presented at the upper left hand corner.**

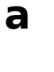

**Figure 2: a) Map of the mean current values derived from ADCP, glider, HF radar and altimetry data over the period 03/2013 – 10/2014. b) Zoom in the northern Ligurian Sea (black rectangle indicated in Figure 2a). The 200-m (red line) and 1000-m (black line) are also shown. Current values are positive (negative) to the right (left) of the ship, glider or satellite tracks.**

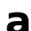

**Figure 3: a) Map of the standard deviations of the velocities derived from ADCP, glider, HF radar and altimetry data over the period 03/2013 – 10/2014. b) Zoom in the northern Ligurian Sea (black rectangle indicated in Figure 3a). The 200-m (red line) and 1000-m (black line) are also shown**

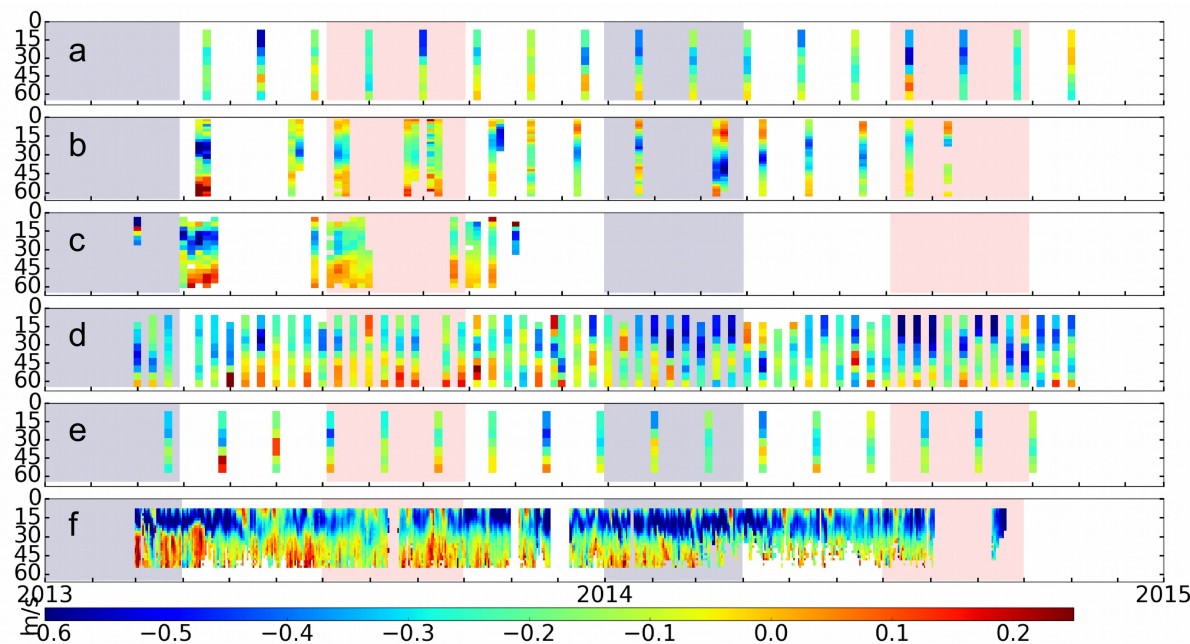

**Figure 4: Time-space diagrams of the current velocities derived from a) SARAL track 887 b) ADCP, c) Gliders, d) Jason 2 track 222, e) SARAL track 302 and f) HF radars between March 2013 and October 2014. The pink and purple areas in the background of the diagrams correspond to the summer and winter seasons, respectively.**

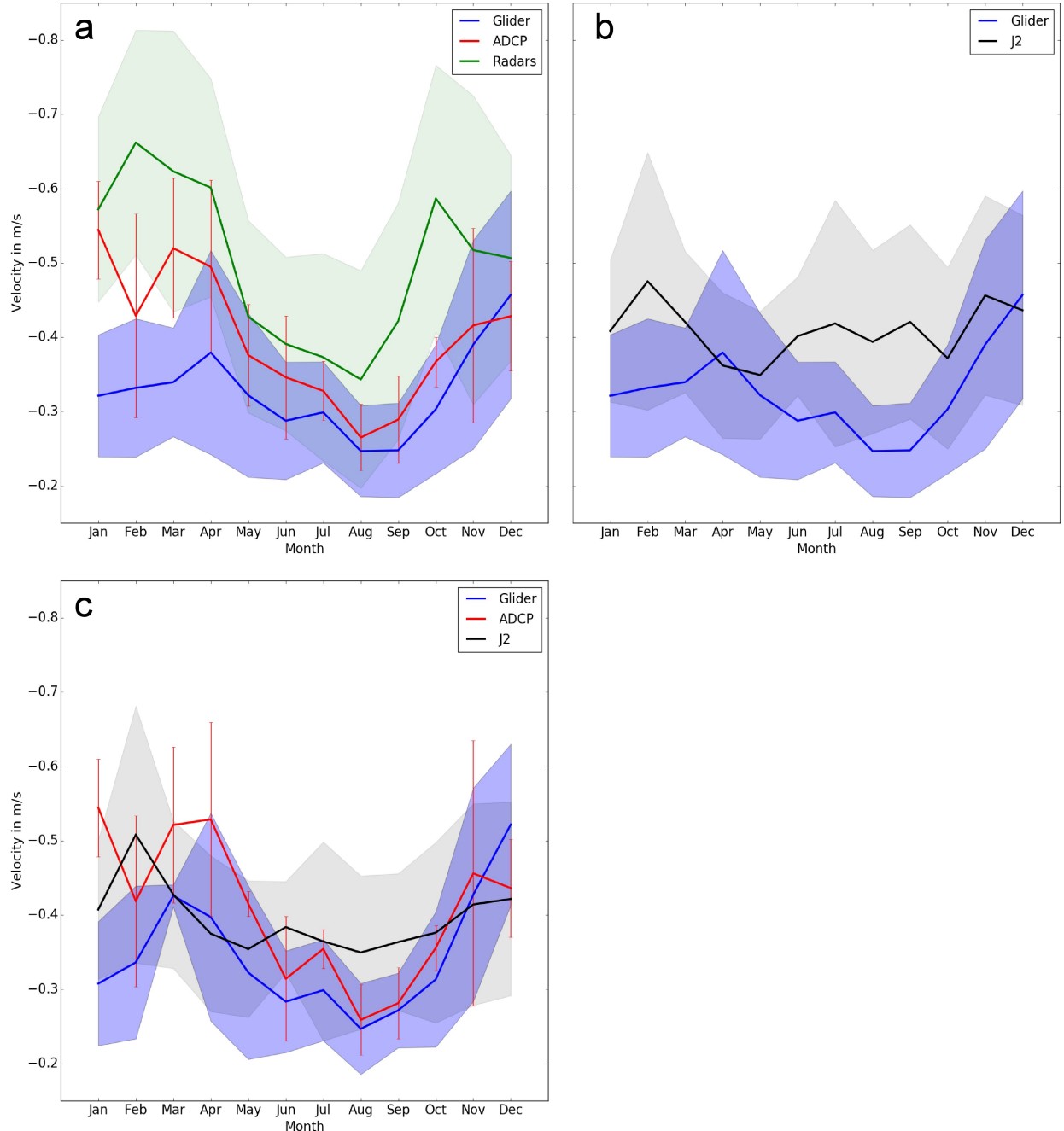

**Figure 5: Seasonal variations of the maximum current amplitude derived from the a) HF radars (green line), ADCP (red line), gliders (blue line), and b) Jason 2 (black line) and glider (blue line) observations available over the period 01/01/2010 - 31/12/2016. c) Same than a) and b) but computed over the period July 2008 to June 2014 and only for the gliders, ADCP and Jason 2. For all the curves the monthly standard deviation of the maximum current amplitude derived from the corresponding instrument is also indicated (curve envelopes and error bars)**

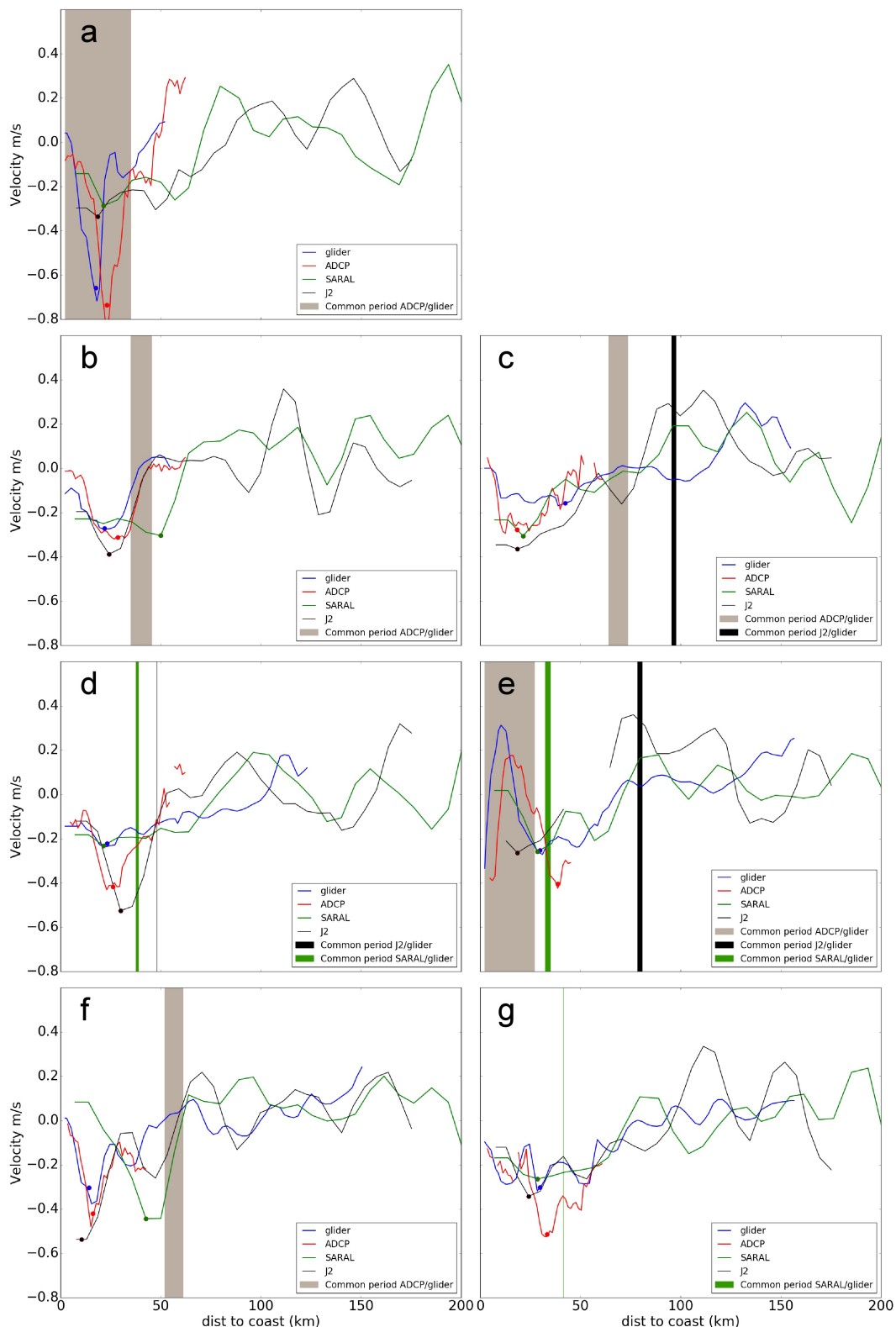

**Figure 6: Cross-shore sections of currents deduced from the glider (blue), ADCP (red), SARAL (green) and J2 (black) altimetry data for the 7 individual cases identified in Table 3. Overlapping periods between the different observations are also indicated.**

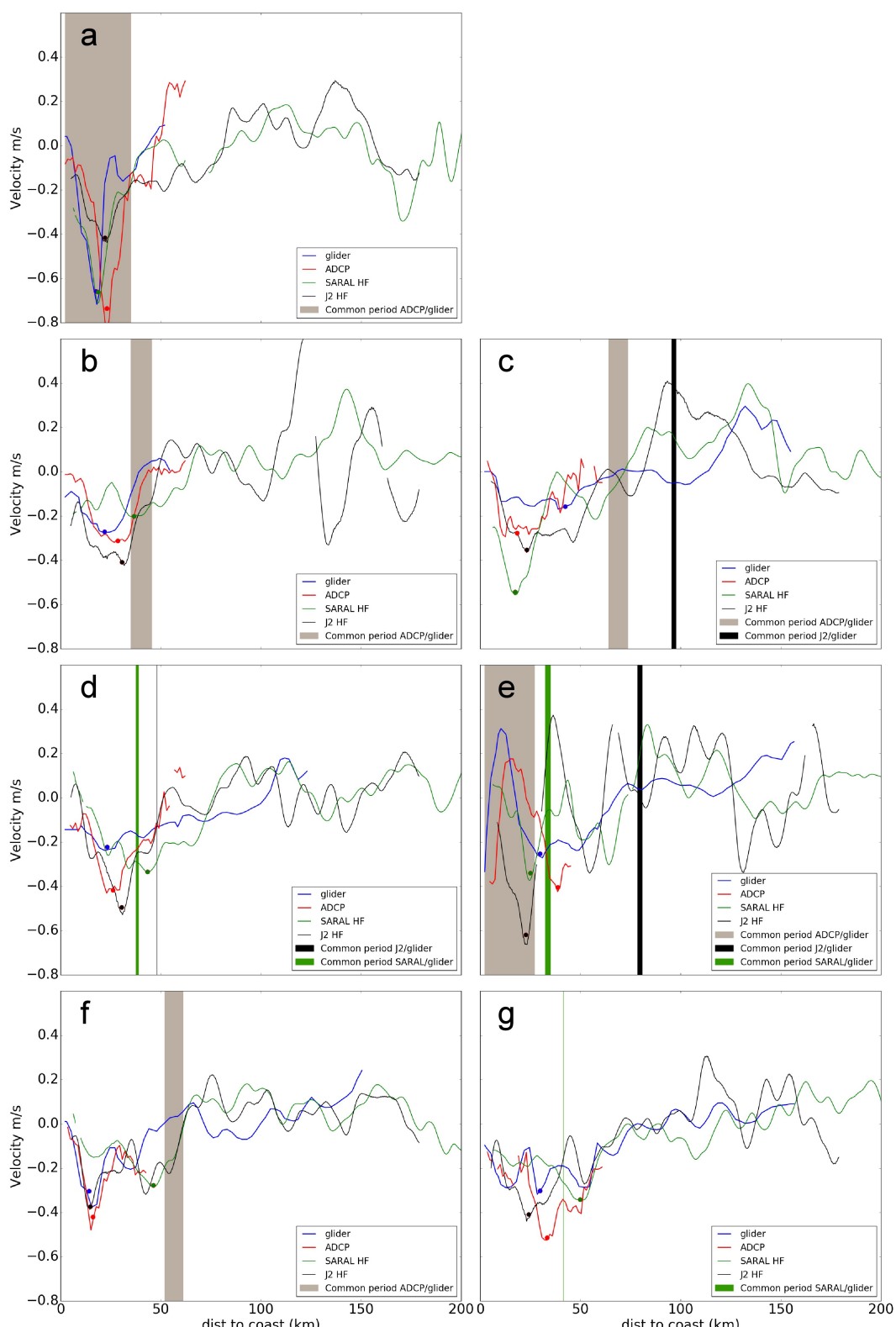

**Figure 7: Same as Figure 6 but for HF altimetry data**

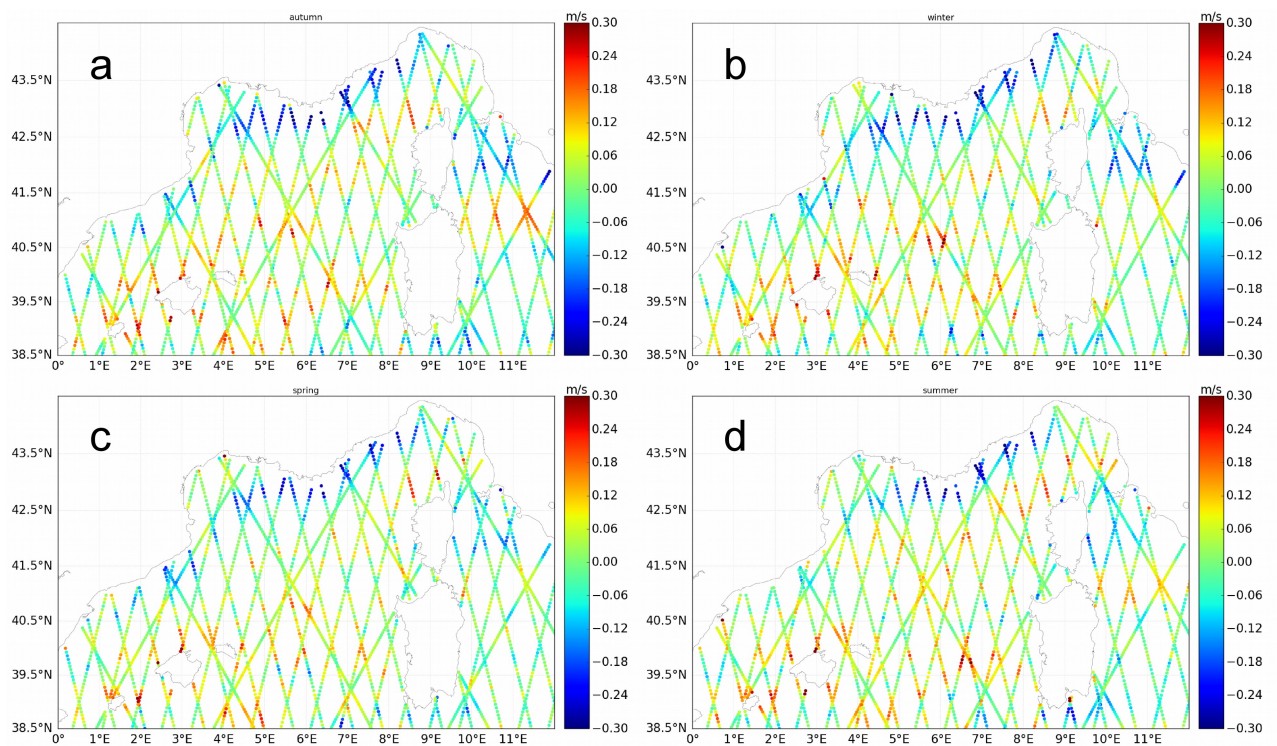

**Figure 8: Seasonal climatology maps of cross-track geostrophic currents (in m/s) derived from Jason 2 and SARAL/AltiKa altimeter data over the period April 2013 – April 2016.**