# Peer review of "Synergy between in situ and altimetry data to observe and study the Northern Current variations (NW Mediterranean Sea)."

_Ocean Science, 2018_

## Referee Comment (RC1) · Anonymous Referee #1 · 18 Aug 2018

The work presented by Carret et al compares SARAL and Jason 2 altimetry data with HF radar, vessel-mounted ADCP and glider data in the North Western Mediterranean Sea. Dataset are well presented, and a complete Section describes similitudes and differences between the dataset. Yet some important physical differences between the dataset are missing in that section. Most of the results presented are not new: it is well known that seasonal and mean average of the altimetry currents are trustable. Also, little is learnt in terms of description of the currents. On the other hand, it is presented a very interesting description of six cases where a detailed comparison between the datasets is made. I encourage the authors to push forward the analysis of the higher frequency and to clearly show when satellite altimetry works well and when does not.

[Figure]

I hope that the below specific comments will help in that sense.

Specific comments:

P3, L 13, add a comma after "swaths"

P3, L29-30, the list of articles is non-exhaustive. Please add "eg" at the beginning of the list.

P4, L 14: "associated to important mesoscale and sub-mesoscale variability at all time scales". Meso and sub-mesoscale have time scales associated as well. Please re-phrase.

P6, L3-5: could you comment why the optimal spatial filtering scale that you obtained is so different for tracks that are relatively close to each other?

P6, L7: Please justify the values used (for SARAL in the precedent paragraph you obtained values that ranged between 34km and 49km). Why you selected a fixed value?

P6, L13: please add a short discussion (with references) to justify that the selection of the MDT. An inaccurate MDT can largely bias your results.

P6, eq 1: it should be noted that this is the across-track component of the geostrophic velocities

P6, L29-30: how much is "too far away" and "too short" ?

P6, L31-P7L1: please improve sentences (for instance obs have the potential...).

P7, L6, a word is missing (end of the line)

P7, L8-9: Here and all over the document: Try to avoid parenthesis as much as possible

P7, L13, "of the second order" - > "of second order"

P7, L15, add "data" after "salinity"

P7: 15-18: please clarify that these geostrophic velocities do not represent the same physical quantity that the ones obtained from satellite altimetry

P8: HF-radar: please add a sentence explaining the error associated to this dataset (ie explaining where velocity components are better solved in the spatial domain covered by the antennas)

P8, L24: altimetry currents are not "located at the surface". They are computed from the SSH, but the SSH topography is the result of several process, including the density changes in the whole water column. Comparison of currents from different instruments elsewhere show that satellite altimetry represents better sub-surface than surface currents. Depth of best matching depends on time and space.

P9, L33-34: gliders provide density sections from where you can extract only the baroclinic component of the velocities. Altimeters provide SLA. When adding MDT, altimetry provides barotropic and baroclinic components. Depending on the accuracy of the geophysical corrections, altimetry data might be more or less biased by ageostrophic components. Please state more clearly the differences between gliders and altimetry data.

P10, L1-4: exactly what I expressed above for comment in P8, L24.

Figure 2: representation of mean velocities for the HF radar could be improved. There, you can solve two directions. The large blue spot is not very meaningful.

P10, L31-32: this information should be included in the legend of the Figure.

Figure 4: please add monthly ticks in the x axis. Please describe how HF radar data were treated. You averaged them along the coast? If so, please discuss how much variability is lost, as the distance along the coast is not so short.

Figure 4f: some interannual variability is also observed. And during 2014 some noisy(?) data close to the coast are also observed. Why it is observed only during that period of time?

Track 302 of SARAL is particularly suited to compare with the HF Radar dataset. Did you try different re-tracking procedures (ALES for example) to analyze how close to the coast the altimetry data can be improved?

P13, L35 & P14, L1-2: please provide a clearer explanation on the criterium adopted.

P14, L2-4: which velocity is seasonally averaged? Legend of Figure 5 says "maximum current amplitude" but form the text I understand that all velocities have been averaged

P14, L4-6, please improve sentences.

P14, L18: South of Toulon only SARAL data can be compared to HF radar data. Please add Toulon position in Figure 1.

P16, L11-13. Please justify the window time scales selected. I suggest to repeat the calculation as a function of the time window. In the coastal region time scales are shorter than 22 days.

P16, L19-22. Figure 6a. I wonder how the distance to the coast is measured. Figure 1 clearly shows that there are no measures of the altimeter inside the 1000m isobath, while gliders and ADCP do show measures up to the 200m isobath. Thus, I am suggesting that in Figure 6 Saral and J2 lines are not correctly placed. Orientation of J2_0009 track is quite different from Saral_887 (with respect to main direction of the isobaths).

P16, L34 to P17, L4: data are "very close in time" but then you argue that differences may be due to "one-week difference". Please say precisely what is the difference in time for each case.

Figure 7 looks strange: double colorbars? Double x-axis?

P20, L2-4 "but a quantification of the high frequency component of the coastal ocean dynamics that altimetry is able to capture would require data that are colocalized in both space and time." Completely agree. But with the dataset that you already have,

you do have the possibility to quantify this quite precisely: how much is the bias that is introduced in the comparison because of non-colocalized data? Just compare, more precisely than what you have done so far, the "very close" space & time datasets with the "not very close".

Discussion can be shortened and concentrate/highlight more on what the results of the work show.

---

## Referee Comment (RC2) · Anonymous Referee #2 · 19 Sep 2018

In this manuscript, the authors use 1 Hz along track coastal altimetry from Jason-2/AltiKa in synergy with other data sources (HF radar, gliders, ship-mounted ADCP) to characterize the variability of the Northern Current flowing along Liguria and French coasts, in particular of the mean flow and seasonal cycles. The authors made a comparatively detailed analysis of the various data sets and described agreements/disagreements in the observed variability and the complementary contribution of each data set to advance the understanding at different scales. In addition to a regional view, the authors provide six case-studies at particular dates. This is a potentially worthwhile study to better understand the full capability of coastal altimetry and at what extent it can be used closer to the coast. This study has the potential to make

a significant contribution, but I have two important remarks:

1) What is the novelty of this paper compared to previous papers that used the same CTOH 1 Hz along track coastal altimetry and focused on the same variability scales (i.e., mean, seasonal and inter-annual flow ?). The paper is not clearly explaining the scientific advance in terms of understanding Northern Current. I have the feeling that the paper describes data very well, but not answering scientific questions related to Northern Current dynamics. My recommendation is to reinforce the discussion (at present it looks like a summary) elaborating major findings in the context of the existing bibliography related to Northern current (from in situ, modelling, altimetry and other remote sensing studies).

2) It is well known that the 1 Hz (7 km) sampling in the coastal zone limits the exploitation. There are many papers that show clearly that longer temporal scales are well reproduced and that the actual challenging in coastal altimetry is to dig the finer ocean scales (along track) and cross-track merging missions (there are actually six missions flying at same time). We have now SAR mode providing improved native along track spatial resolution and better signal to noise ratio. We have retracked data for conventional missions that push resolution at 20 Hz. All these innovations are very promising to study high frequency mesoscale. For example, AltiKA has native resolution at 40 Hz, why reducing to 1 Hz ? I really recommend the authors to investigate data at higher sampling rate as this would be a really step in advance. Therefore, my position is major review as the results are not new, but potentially to become of high interest to the oceanographic community if authors re-focus the analysis on high resolution altimetry

Hereinafter additional comments:

Pg. 3, line 1, "Radar altimeters measure the sea surface height (SSH) variations": the sentence is uncorrect. The radar altimeter transmit pulses. The system measures the time pulses take to be reflected back satellite. Time is then converted in distance, corrected for various effects and then referred to earth using orbit altitude (this is the

so called SSH). Please be precise

Pg. 3, line 2, "ever more accurate observations": more accurate than what? Please clarify

Pg. 3, line 9, "the first global synthetic aperture radar (SAR, or Delay-Doppler) technique": the term "SAR" has to be sued properly to avoid confusion. The reader would understand the first global SAR. SAR, also known as Delay Doppler, is a new coherent processing mode of individual echoes (conventional altimetry uses incoherently processing). Please better clarify

Pg. 3, line 10, "enhanced accuracy and reduced noise": the use of the term "accuracy is wrong". Reduced noise means better precision. The other enhancement is increased native along track resolution

Pg. 3, line 13, "the SWOT . . .a new step forward": reference missing

Pg. 3, lines 14-15, "particularly important to monitor the sea level variations, directly related to our living environments and marine ecosystems": This sentence is too vague. You have to better explain (in term of processes, e.g. flooding, etc.) why sea level changes are more important in the coastal zone than deep waters.

Pg. 3, line 16, "from these new altimetry techniques": The statement is vague. Maybe use the term "modern altimetry" to include all technical improvements (Delay Doppler processing, small footprint) as well as reprocessed conventional altimetry (retracking, new/improved corrections, etc.).

Pg. 3, lines 17-18, "The strongest limitation is the modification of the radar echo in the vicinity of land": here the reader might thing that the problem is only land contamination. Indeed, there are other effects , e.g. inhomogeneity of the water surface (Brown altimetry assumes homogeneous scattering). The authors have to be more rigorous here citing proper literature

Pg. 3, line 27, "new altimetry techniques are intrinsically less sensitive to the land

contamination": The statement is not correct. Again the term "techniques" is not appropriate. Moreover, land contamination cannot be used in general sense., e.g. SAR mode has few effect if the track is parallel to land.

Pg. 1, line 28, "They provide more robust and accurate measurements, ever closer to the coast": The athors have to provide figures about accuracy as function of distance with concrete examples from bibliography

Pg. 3, lines 29-30, "We can easily predict that the use of altimetry in coastal studies": the reader here expects illustrating major findings from these studies (i.e. state-of-the-art)

Pg. 3, lines 32-33, "Coastal observations are mainly based on in situ instruments and satellite imagery (sea surface temperature and ocean color images): I don't understand this sentence. The coastal observing system is multi-sensor, multi-platform. SAR imagery is especially useful in the coastal zone due to its high spatial resolution. The sentence has to be reworded.

Pg. 4, lines 1-2, "in a growing number of regions": Please provide examples

Pg. 4, line 2, "in conjunction": better using in "synergy"

Pg. 4, lines 4-5, "in situ observations cover more limited areas and often provide time series that present large gaps.": please provide examples. What means gaps in time series ? are you referring to tide gauges ? buoys? Please better clarify

Pg. 4, lines 6-7, "satellite imagery is often impacted by clouds and does not provide any direct information on the changes occurring in the water column.": the sentence is confusing the reader. Clouds might be a problem for optical imagery, but not for microwave sensors (e.g. scatterometry, SAR). Moreover, although satellites maps the ocean surface, it si possible to derive info in the water columns (one example is SAR detecting internal waves)

Pg. 4, line 8, "almost-global synoptic observations". Satellite altimetry provide global

coverage. Revisting the same place depends on the mission (e.g. Cryosat is drifting in orbits and revisits the same place in along time). What about "synoptic" ? e.g. Jason takes 10 days to get a global coverage. What do you mean with "almost"? – the sentence in unclear. The reader might be confused, e.g. an optical imagery is synoptic. Infor at all pixels is at same time)

Pg. 4, line 11, "conjunction": use "synergy"

Pg. 4, line 13, "ideal": use another word (e.g. laboratory and explain why in detail)

Pg. 4, line 14, "at all time scales"; clarify which scales (i.e. range).

Pg. 4, line 20, mesoscale variability is higher in autumn and winter": Why higher in these seasons ? please explain

Pg. 4, line 25, "to partially capture": please explain why "partially"

Pg. 4, line 26, "to provide original aspects of the regional circulation": please explain "original aspects"

Pg. 4, line 27, "coherent circulation patterns": please illustrate these patterns

Pg. 4, line 29, "found similarities": which ones?

Pg. 4, line 33, "compared to the other coastal ocean observing systems": please specify which coastal observing systems

Pg. 4, line 34, "Ligurian Sea": better to use Ligurian-Provencal basin, because HF radars are not in the Ligurian Sea (as it is usually defined in term of boundaries)

Pg. 5, line 15, "The performance of SARAL is significantly better than Jason-2": please provide references stating that with figures

Pg. 5, line 19, "SARAL tracks 302, 343 and 887": why not using also the other tracks ?

Pg. 5, line 23, "Sea Level Anomalies (SLA) every 6-7 km: I am but surprised authors use this low along track sampling (7 km). As the novel aspect is the finer scale of ocean

circulation, the authors have to use the high res altimetry (350 m) that are reprocessed using re-tracking.

Pg. 6, line 2, "in Morrow et al., 2017, the data located over the continental shelf were discarded": this is further point that support the need of using high res altimetry

Pg. 6, lines 1-4, "We did ….39 km for the SARAL track 343, 34 km for the SARAL track 887 and 67 km for Jason 2 track 222.. altimetry observations": I see that Morrow et al., 2017 used these figures come from standard along-track data at 1 Hz (7 km) discarding data in the coastal zone. They found some figures about size of structures and optimal filtering. The authors here use reprocessed along-track data at 1 Hz with no retracking applied, but with more data coverage going to the coast. They found lower figures about size of structures. I am confused as we talk about average values and sampling is not changing. Is signal-to-noise better ? you have to demonstrate, because structures can emerge from background noise only the ratio is higher . I think the scale would only change (become finer) only if authors enhance resolution of their altimeter data. Moreover, the author do not explain why scales vary with close tracks.

Pg. 6, line 8, "35 km for SARAL": why do you set 35 km if tracks have different scales? Please justify

Pg. 6, line 13, "from (Rio et al.,2014)": change to "Rio et al., (2014)" Moreover, the authors have to demonstrate that this MDT is accurate going closer to the coast as in open ocean (this product was not generated to be used in the coastal zone)

Pg. 6, lines 29-30 "The ones being too short or moving too far away from an average trajectory": please be rigorous in stating "too short" and "too far away"

Pg. 7, line 6, "points and the was too": typo to correct

Pg. 7, lines 12-13, "compare the currents derived from these data with the currents measured or derived from the other instruments": becareful that altimeter derived currents from altimetry are not equivalent to currents measured e.g. from ADCPs

Pg. 8, line 17, "HF radar is roughly 60x40": is this the area covered? How much is accuracy of currents? Please discuss bibliography

Pg. 9, lines 4-5, "to altimetry data still remains limited in the 10- 15 km coastal band" The statement is wrong. With reprocessed high res altimetry tat adopts retracking one can go closer to the coast.

Pg. 9, line 19, "HF radars and altimeters observe the ocean surface": altimetry provides geostrophic currents that are derived from SLA where tides and atmospheric effects (wind and pressure) are removed. HF radar provides the real total current at surface. . Also gliders provide only the baroclinic component of currents. ADCP measure currents at differet layers. Authors have to discuss in detail these differences.

Pg. 9, lines 33-34, "the gliders and altimeters are clearly the closest in terms of current information derived.". I don't agree. Gliders miss the barotropic component due to atmospheric effects that in the coastal zone is not negligible.

Pg. 10, line 22, "from March 2013 to October 2014": I am not sure this is a good approach. Mean flows have sense if you average by month, season or annual.

Pg. 11, lines 1-2, "Fig. 2, where one can notice a very good consistency of the mean currents derived from all the different instruments."

Pg. 11, lines 32-33, "HF radars ($\sim$-0.44 m/s) than in altimetry ($\sim$-0.29 m/s)": Why this difference ? please explain.

Pg. 12, line 1, "we observe values of 0.12-0.2 m/s": Mean values are around 0.3 m/s and variability is of same order of magnitude (more or less). Is this picture confirmed by bibliography ?

Pg. 13, lines 29-30, "The maximum NC current amplitude is defined as the average of the first decile of the velocity values for each transect and time": please justify this definition

---

## Author Response (AR1)

Alice Carret
Laboratoire d'Etudes en Géophysique et Océanographie Spatiale (LEGOS)
OMP
14 Avenue Edouard Belin
31400 Toulouse
France
Tel : 00-33-5-61-33-26-81
Email : alice.carret@legos.obs-mip.fr

Subject : Manuscript os-2018-76 revision

14 December, 2018

Ocean Science

Dear Editor,

Please find enclosed the revised version of our manuscript entitled "Synergy between in situ and altimetry data to observe and study the Northern Current variations (NW Mediterranean Sea)" , by A. Carret and co-authors.

We are grateful that the reviewers gave us the opportunity to improve the earlier version of the manuscript. The paper has been significantly modified to take the remarks into account. In the following pages we detail the answer to the comments made by the reviewers. We have also included a marked-up manuscript version.

We hope that the revised paper is in an acceptable form now. We look forward to your decision.

Best regards,

Alice Carret.

First, we would like to thank the two reviewers for providing constructive comments that were taken into account in order to improve the manuscript. Please find below a point-by-point answer. The reviewers comments are in bold.

**Reply to reviewer #1**

**The work presented by Carret et al compares SARAL and Jason 2 altimetry data with HF radar, vessel-mounted ADCP and glider data in the North Western Mediterranean Sea. Dataset are well presented, and a complete Section describes similitudes and differences between the dataset. Yet some important physical differences between the dataset are missing in that section. Most of the results presented are not new: it is well known that seasonal and mean average of the altimetry currents are trustable. Also, little is learnt in terms of description of the currents. On the other hand, it is presented a very interesting description of six cases where a detailed comparison between the datasets is made. I encourage the authors to push forward the analysis of the higher frequency and to clearly show when satellite altimetry works well and when does not. I hope that the below specific comments will help in that sense.**

Thank you for these comments. First, we improved the description of the differences between the datasets by adding in Section 2 more informations and/or precisions on their respective physical content (details below). We also used high resolution SLA altimetry data (i.e. 20Hz for Jason-2, 40-Hz for SARAL) and added a section (new section 3.4) to compare and discuss the corresponding results with the results obtained from 1-Hz SLA data. We then insisted on the individual cases as suggested by the reviewer. However we believe that the results presented in this manuscript, even in its first submitted version, are really new. If different studies have already shown that altimetry is able to capture seasonal and mean currents, they did not (or poorly) show until what point this is true and what part of the seasonal and mean current components is missing. Past studies on coastal currents derived from altimetry are generally qualitative (as in Birol et al., 2010-2014-2015; Jébri et al, 2016) and/or based on individual case studies. Here we take advantage of a large number of data (as much as we found) and relatively long time series in order to quantify (as much as we can) what part of the current can/can't be captured by altimetry. We used a multi-platform approach in order to learn more on the causes of the differences between currents derived from altimetry and from in situ data. To our knowledge, this is not common at all since we do not know coastal altimetry studies based on such degree of integrated observation system.

**P3, L 13, add a comma after "swaths"**

It has been corrected.

**P3, L29-30, the list of articles is non-exhaustive. Please add "eg" at the beginning of the list.**

Right. It has been done.

**P4, L 14: "associated to important mesoscale and sub-mesoscale variability at all time scales".Meso and sub-mesoscale have time scales associated as well. Please re-phrase.**

We have removed these words. Now : "To study the contribution of altimetry amongst other types of coastal ocean measurements, the North-Western Mediterranean Sea (NWMed) represents a laboratory area. First, with a Rossby radius of only ~10 km, the region is associated to a variety of mesoscale and sub-mesoscale dynamical signals (see below)."

**P6, L3-5: could you comment why the optimal spatial filtering scale that you obtained is so different for tracks that are relatively close to each other ?**

We agree and have added the following sentences at the end of the paragraph: "The lower values obtained for  SARAL are due to the better signal-over-noise ratio of the AltiKa altimeter, compared to Jason-2. The differences obtained between the three SARAL tracks are explained by their respective geographical locations: they represent different mesoscale features.".

**P6, L7: Please justify the values used (for SARAL in the precedent paragraph you obtained values that ranged between 34km and 49km). Why you selected a fixed value?**

We have added the following sentence: "Note that we have chosen a single value for the different SARAL tracks in order to have the same data processing and facilitate the comparison between the different datasets".

**P6, L13: please add a short discussion (with references) to justify that the selection of the MDT. An inaccurate MDT can largely bias your results.**

We agree. We chose to work with the regional MDT from Rio et al., 2014 which was validated against in situ datasets. Compared to the previous MDT from Rio et al., 2007, it has a better resolution (1/16° vs 1/8 °) and the regional circulation is better resolved (see Rio et al, 2014 but we have also done our own diagnostics). We have added the following sentences:

"The MDT product used is a regional product with an horizontal resolution of 1/16° (lower than the altimetry resolution in the along-track direction). Compared to other products, it allows a better representation of the NC in the Ligurian Sea (Rio et al., 2014)."

**P6, eq 1: it should be noted that this is the across-track component of the geostrophic velocities**

It has been written.

**P6, L29-30: how much is "too far away" and "too short" ?**

Right. Now: "The ones being too short (<60 km) or moving too far away (>15 km) from an average trajectory computed from the individual ones were discarded. "

**P6, L31-P7L1: please improve sentences (for instance obs have the potential. )**

We have rephrased. Now: "It represents a huge amount of observations and a large number of cases available for the comparisons with altimetry or with the other in-situ observations. "

**P7, L6, a word is missing (end of the line)**

We have added the word "horizontal" in the text.

**P7, L8-9: Here and all over the document: Try to avoid parenthesis as much as possible**

We removed these parenthesis and some others.

**P7, L13, "of the second order" - > "of second order"**

Done.

**P7, L15, add "data" after "salinity"**

Done.

**P7: 15-18: please clarify that these geostrophic velocities do not represent the same physical quantity that the ones obtained from satellite altimetry**

We have added the following sentence: "The difference with altimetry-derived currents is then that the barotropic component and the baroclinic component below 500m are missing."

**P8: HF-radar: please add a sentence explaining the error associated to this dataset (ie explaining where velocity components are better solved in the spatial domain covered by the antennas)**

We have added the following sentence : "An assesment of this HF Radar site can be found in Sentchev et al. (2017) who found an overall good agreement between derived radial velocities and in situ ADCP, with relative errors of 1 and 9 % and root mean square (RMS) differences of 0.02 and 0.04 m/s, slightly increased, in velocity and direction, for the reconstructed total velocities, but mainly in conditions of unstationnary wind forcing. »

**P9, L24: altimetry currents are not "located at the surface". They are computed from the SSH, but the SSH topography is the result of several process, including the density changes in the whole water column. Comparison of currents from different instruments elsewhere show that satellite altimetry represents better sub-surface than surface currents. Depth of best matching depends on time and space.**

It has been reworded. Now: " We then decided to use the glider data at 34 m depth (to be coherent with the ADCP observations) and consider that it should not be a significant source of differences with altimetry currents, representing near-surface currents"

**P9, L33-34: gliders provide density sections from where you can extract only the baroclinic component of the velocities. Altimeters provide SLA. When adding MDT, altimetry provides barotropic and baroclinic components. Depending on the accuracy of the geophysical corrections, altimetry data might be more or less biased by ageostrophic components. Please state more clearly the differences between gliders and altimetry data.**

Done. Now: "After the addition of the MDT, the gliders and altimeters are clearly the closest in terms of current information derived. However, the glider currents are computed from hydrographic measurement profiles with a reference level of 500 m. They miss the barotropic and the deeper baroclinic geostrophic current components when altimetry and MDT allow to estimate absolute geostrophic currents representative of the horizontal density gradients integrated over the whole water column. In this study, in order to minimize (as far as possible) the differences between the current data sets, we performed a projection of the ADCP velocities to obtain the current component perpendicular to the ship transects. Concerning the gliders, estimates of depth-average currents computed following *Testor et al*., 2018 approach were added to the velocity data as an approximation of the barotropic component."

**P10, L1-4: exactly what I expressed above for comment in P9, L24**.

**Figure 2: representation of mean velocities for the HF radar could be improved. There, you can solve two directions. The large blue spot is not very meaningful.**

We have chosen to represent only the zonal component to be closer to the information which can be derived from the other data sets. However in this area the NC is known to be almost zonal. These informations were missing in the text and have been added: "Concerning the HF radars, only the zonal current component is taken into account. Note however that in this area, since the NC is almost zonal, most of its mean and variability are captured in the corresponding statistics." The representation of the two direction overload the figure and we have then decided not to change.

**P10, L31-32: this information should be included in the legend of the Figure**

This sentence has been moved in the legend of Figure 2.

**Figure 4: please add monthly ticks in the x axis. Please describe how HF radar data were treated. You averaged them along the coast? If so, please discuss how much variability is lost, as the distance along the coast is not so short. →**

We have added monthly ticks in the x axis and we have added the following sentence: "The HF radar data correspond to a meridional section of the zonal current component located at 6.2°E." See also answer to the comment on Figure 2.

**Figure 4f: some interannual variability is also observed. And during 2014 some noisy(?) data close to the coast are also observed. Why it is observed only during that period of time?**

We have no clear explanation for the presence of the noisy HF radar data located close to the coast in 2014 (these data are processed and distributed by MOOSE) and have decided to remove these points.

**Track 302 of SARAL is particularly suited to compare with the HF Radar dataset. Did you try different re-tracking procedures (ALES for example) to analyze how close to the coast the altimetry data can be improved?**

No we did not try yet but it will be done in the near future. We wait for the new 20-Hz L3 ALES/X-TRACK product which should be distributed soon.

**P13, L35 & P14, L1-2: please provide a clearer explanation on the criterium adopted.**

Now: "The maximum NC current amplitude is defined as the average of the first decile of the velocity values for each transect and time (remember that the NC corresponds to negative current values). These values must be close in space. This strategy allows to filter large isolated current values which may not correspond to the NC. In altimetry, only a distance spanning 60 km to the coast is considered. The number of data in the first decile varies according to the data set and to the number of data in the section considered (because of the lower resolution, it always corresponds to one point in altimetry). As we can see in Figure 4d, data gaps exist in Jason 2 for some cycles. When more than 3 points are missing, the corresponding cycle is discarded from the analysis."

**P14, L2-4: which velocity is seasonally averaged? Legend of Figure 5 says "maximum current amplitude" but form the text I understand that all velocities have been averaged**

It has been reworded. Now: "Finally, all the maximum NC current values collected are averaged ..."

**P14, L4-6, please improve sentences.** →

Now: "The results derived from in-situ data are in Figure 5a and the results derived from altimetry are in Figure 5b. The glider results are on both figures because this instrument provides the currents which are the closest to altimetry in terms of physical content."

**P14, L18: South of Toulon only SARAL data can be compared to HF radar data. Please add Toulon position in Figure 1**

Done.

**P16, L11-13. Please justify the window time scales selected. I suggest to repeat the calculation as a function of the time window. In the coastal region time scales are shorter than 22 days.**

Please see the answer to the comment below (p 20, L 2-4).

**P16, L19-22. Figure 6a. I wonder how the distance to the coast is measured. Figure 1 clearly shows that there are no measures of the altimeter inside the 1000m isobath, while gliders and ADCP do show measures up to the 200m isobath. Thus, I am suggesting that in Figure 6 Saral and J2 lines are not correctly placed. Orientation of J2_0009 track is quite different from Saral_887 (with respect to main direction of the isobaths).**

Right. The figures are not represented as a function of the distance to the coast but of the distance to the transect-shoreline intersection point. The new sentence is: "The corresponding cross-track currents are shown in Figure 6 (by season) as a function of the distance to the point where the corresponding transect intersects the coastline."

**P16, L34 to P17, L4: data are "very close in time" but then you argue that differences may be due to "one-week difference". Please say precisely what is the difference in time for each case.**

All the dates are provided in Table 3 for each case and each instrument. We have added this information in the corresponding sentence. Now: "It could be due either to the differences in the dates of observations (one week from Table 3, temporal scale at which meanders develop) or to an important ageostrophic NC component."

**Figure 7 looks strange: double colorbars? Double x-axis?**

The figure was inserted twice. It has been corrected.

**P20, L2-4 "but a quantification of the high frequency component of the coastal ocean dynamics that altimetry is able to capture would require data that are colocalized in both space and time." Completely agree. But with the dataset that you already have, you do have the possibility to quantify this quite precisely: how much is the bias that is introduced in the comparison because of non-colocalized data? Just compare, more precisely than what you have done so far, the "very close" space & time datasets with the "not very close".**

Thank you for this suggestion. We have tried to investigate the bias introduced by non-colocalized data in more details and have computed the diagnostics shown below. Using all the data available, we represent in Figure 1 and in Figure 2 the differences between the maximum NC amplitudes derived from in situ datasets (gliders and ADCP) and from altimetry (J2 and SARAL) as a function of the number of days which separates two measurements. In Figure 3 and 4 we represent the differences obtained as a function of the distance of the NC core to the coast. As you can see, unfortunately, these results don't allow to draw any conclusion because there are no clear rule that appears. The explanation is not really obvious. Is it because of the high level of short scale variability in this area? Difficult to say. A high resolution numerical model would probably help to understand but it is beyond the scope of this study.

[Figure]

[Figure]

*Figure 1: Differences of the maximum amplitudes of the NC between the ADCP and SARAL (in red) and between the ADCP and J2 (in black) in function of the time difference*

*Figure 2: Differences of the maximum amplitudes of the NC between the gliders and SARAL (in red) and between the gliders and J2 (in black) in function of the time difference*

[Figure]

[Figure]

*Figure 3: Differences of the maximum amplitudes of the NC between the ADCP and SARAL (in red) and between the ADCP and J2 (in black) in function of the NC distance to coast obtained from the ADCP*

*Figure 4: Differences of the maximum amplitudes of the NC between the gliders and SARAL (in red) and between the gliders and J2 (in black) in function of the NC distance to coast obtained from the gliders*

**Reply to reviewer #2**

**1) What is the novelty of this paper compared to previous papers that used the same CTOH 1 Hz along track coastal altimetry and focused on the same variability scales (i.e., mean, seasonal and inter-annual flow ?). The paper is not clearly explaining the scientific advance in terms of understanding Northern Current. I have the feeling that the paper describes data very well, but not answering scientific questions related to Northern Current dynamics. My recommendation is to reinforce the discussion (at present it looks like a summary) elaborating major findings in the context of the existing bibliography related to Northern current (from in situ, modelling, altimetry and other remote sensing studies).**

We partly agree and have tried to clarify the objective of this paper relative to previous work in a new version of the introduction (see below). This paper does not focus on scientific advances concerning the NC. This is the next step but, from our point of view, the priority is still to promote the use of altimetry data for coastal circulation studies (not really used outside of the small community of coastal altimetry experts). Therefore we need to better demonstrate and quantify what can be observed by altimetry in terms of coastal current variability. In this paper we take the opportunity of a large number of data from a variety of plaforms, that are commonly used in coastal research, to perform systematic cross-comparisons. To our knowledge it is the first time that this type of work is done, and with such integrated approach. From this study we demonstrate that in average 1-Hz altimetry data (corresponding to the standard products available for the scientific community) are able to capture 50% of the NC variability. But for individual cases, this number varies a lot from one situation to the other (and not only as a function of the season). We think that these results are important. To promote the use of a given instrument, to show and quantify what can be observed and what can not is a very important issue. Note that we also decided to add high sampling rate data (see the answer to the next comment) to the analysis, even if the results are not so clear and if no consensus exists yet on the way to process these data (that means that only experimental products exist and that they are rarely used). However, this multi platform study also provides new informations on the NC system variability (even if it is not the main focus of the study). It reveals a strong NC increase in summer 2014 (not expected in this season) and the presence of an eastward current flowing ~50-80 km off the coast in the Ligurian Sea.

Anyway we think that the introduction and the objectives of the study needed indeed to be clarified relative to previous work and it has been done. We have made changes in the following paragraphs :

« To study the contribution of altimetry among other types of coastal ocean measurements, the North-Western Mediterranean Sea (NWMed) represents a laboratory area. First, with a Rossby radius of only ~10 km, the region is associated to a variety of mesoscale and sub-mesoscale dynamical signals (see below). As a result it represents a challenge for altimetry observations. Secondly, the number of in-situ observations is relatively important in this region, allowing comparison with independent data. In the NWMed, the main feature of the

surface ocean circulation is the Northern Current (called NC hereafter) which is formed in the Ligurian Sea (*Taupier-Letage and Millot*, 1986) and flows cyclonically along the Italian, French and Spanish coasts. This current presents a marked seasonal variability, with a maximum amplitude from February to April (*Sammari et al.*, 1995; *Millot*, 1991), and it meanders in a vast range of wavelengths (10-100 km). The mesoscale variability is higher in autumn and winter because of the larger baroclinic instability associated to strong and cold winds(*Alberola et al.*, 1995; *Millot*, 1991). During the last 10 years, the NC has been intensively monitored by a variety of in situ data (moorings, research vessels, gliders, and HF radars) collected from the MOOSE integrated observing system (Mediterranean Ocean Observing System for the Environment). Despite a width of only 30-50 km, through the comparison with ADCP current data located in the Ligurian Sea, *Birol et al.*, 2010 demonstrated that reprocessed altimetry data are able to capture half of the amplitude of the seasonal NC variability. The altimetry currents have then been used to analyze the regional current variability at seasonal scale. In the Balearic Sea, the reliability of altimetry currents has been verified by direct comparison with currents derived from gliders and HF radars (*Bouffard et al.*, 2010; *Pascual et al.;* 2015; *Troupin et al.*, 2015). These case studies showed that altimetry can depict current signals coherent with the other instruments. *Morrow et al.*, 2017 also showed that some of the large scale eddies observed by gliders in the NWMed can be captured by altimetry, in particular by the SARAL mission. A more systematic use of altimetry in regional coastal applications requires a better quantitative assessment of its performance near coastlines, from daily to interannual time scales.

The general objective of this paper is not only to investigate the accuracy of the velocity fields derived from altimetry data next to the coast at different temporal scales , but also to define its contribution from the other coastal ocean observing systems which exist in the region (ship-mounted ADCPs, gliders and HF radars). In this study, we combine all the different available in situ data sets which provide information on currents in the Ligurian-Provençal basin and perform systematic comparisons with currents derived from altimetry at different time-scales. In particular, we analyze how the different available observing techniques capture the NC variability and the coherence/divergence/complementarity of the informations derived. From previous studies, we know that only a small part of the NC variations can be captured by conventional satellite altimetry. Here, we use both Jason-2 and SARAL/AltiKa missions to investigate the progress made from Ku-band to Ka-band altimetry. We also investigate the potential of experimental 20/40-Hz altimetry products to monitor the NC variations, relative to the conventional 1-Hz data. »

**2) It is well known that the 1 Hz (7 km) sampling in the coastal zone limits the exploitation. There are many papers that show clearly that longer temporal scales are well reproduced and that the actual challenging in coastal altimetry is to dig the finer ocean scales (along track) and cross-track merging missions (there are actually six missions flying at same time). We have now SAR mode providing improved native along track spatial resolution and better signal to noise ratio. We have retracked data for**

**conventional missions that push resolution at 20 Hz. All these innovations are very promising to study high frequency mesoscale. For example, AltiKA has native resolution at 40 Hz, why reducing to 1 Hz ? I really recommend the authors to investigate data at higher sampling rate as this would be a really step in advance. Therefore, my position is major review as the results are not new, but potentially to become of high interest to the oceanographic community if authors re-focus the analysis on high resolution altimetry.**

We fully understand this comment but here again we agree only partly. First, as explained above, we believe that it is important to promote products that are now at a mature stage for the coastal scientific community and it is the reason why we have decided to keep 1-Hz data as our focus in this study. Secondly, if different experimental high-rate datasets are generated by coastal altimetry research groups, they are still largely at an experimental stage and no consensus exists on the way they should be processed. More than the sea level (SL) estimates, the resulting current fields and then the NC characteristics, are very sensitive to the strategy followed for data processing (including both retracking and corrections), screening and filtering. Of course it is because they depend on SL gradients which computation is very sensitive to the noise and outliers which are important in high-rate coastal altimetry observations. It is illustrated in the results below. However, because we agree that it is important to gain experience on these new higher resolution datasets, we have decided to add a section 3.4 dedicated to high-sampling rate altimetry. It illustrates their potential, but as you can see below, in terms of scientific advance, the results are relatively mitigated. We have added a new Figure 7 and modified Table 4. We have also make changes in the abstract, sections 1, 2 and 4, accordingly.

**« 3.4 Can we improve the estimation of the NC characteristics with high-rate altimetry compared to 1-Hz data?**

In this section we consider the improvement that is possible to obtain in terms of current derivation with the use of high-rate altimetry measurements, compared to the conventional 1-Hz data used above. Here, we used an experimental version of high-rate X-TRACK SLA data for both Jason-2 and SARAL which original measurements are at 20-Hz and 40-Hz, respectively. Since a lot of erroneous data remained in the coastal area, we applied a 2-sigma filter on the resulting SLA fields along each individual track and cycle, in order to edit the data before filtering and computation of the current estimates (section 2.1).

In order to analyze if we can expect a better observation and understanding of the NC hydrodynamics from high-rate altimetry measurements, they have been used to compute the same diagnostics than in sections 3.1, 3.2 and 3.3. Only the results for the individual snapshots will be illustrated here (Figure 7) since, even if the major difference with the current fields derived from 1-Hz altimetry is that the larger number of coastal data allows to estimate currents closer to the coast and then to better resolve the NC flow (see Figure 7), we did not find significant differences in the NC statistics (mean current and standard deviation values) and amplitude of the seasonal cycle computed from 20/40-Hz SLA.

In Figure 7, the same color code than in Figure 6 is used while, for each case, the maximum NC current amplitude andcorresponding location are reported in Table 4 for both SARAL and Jason-2 (as in section 3.3). In case 1 (Figure 7a), the gain obtained with the use of HF data is very clear. At this date the NC vein is narrow and located near the coast. Contrary to 1-Hz solution, the NC is better resolved by both SARAL and Jason-2 high-rate altimetry. It is especially true for SARAL which NC characteristics are almost identical to the glider ones. In Jason-2, the NC core is close to the glider solution but its amplitude is ~35% lower. For cases 2 and 4 which correspond to the summer (Figure 7b and c), here again the use of high-rate altimetry allows a better observation of a NC vein but the agreement with in-situ data is not so good. Concerning SARAL, for case 2, the current estimates are suspect since a reduction of the current intensity appears at the location of the NC core in the other datasets (Figure 7b). For case 4, SARAL NC amplitude is too high (0.55 m/s vs 0.16 m/s for the glider). Jason-2 high-rate NC estimates appear closer to the in situ data than SARAL for both cases 2 and 4 but, the resulting NC characteristics do not appear better than the corresponding 1-Hz estimations, when compared to ADCP (table 4). It probably reveals that the cut-off frequency chosen in the filtering is too low. Cases 5 and 6 (Figures 7 d and e) also show some very doubtful oscillations in both SARAL and Jason-2 currents and high-rate altimetry does not improve the NC estimations. In winter, the cases 3 and 7 (Figures 7g and f, respectively) are very different. In case 7, 20-Hz Jason 2 data depicts the entire NC with current estimates much closer to in-situ data (especially the glider), compared to 1-Hz Jason 2 measurements, In case 3 they degrade/improve the NC representation (Figure 7g) if we refer to the glider/ADCP, respectively Note that this case illustrates the difficulty of the calibration of altimetry data processing algorithms with independent observations since results may differ as a function of the independent observations used. Here, 40-Hz SARAL data show a too noisy current solution .

In conclusion, as already shown in previous work (*Birol and Delebecque*, 2014; *Gomez-Enri et al.*, 2016), high-rate altimetry allows to derive significantly more sea level data near the coast. Here we observe that the coastal circulation derived is better resolved in space (i.e. both in horizontal resolution and in distance to the coast of the current estimates). But the resulting current fields depend crucially on the strategy followed for data processing (including both retracking and corrections), screening and filtering. 20/40 Hz altimetry data obviously present a clear advantage for coastal studies but the production of these datasets is still at an experimental status and there is room for further improvements."

[Figure]

Figure 7: Cross-shore sections of currents deduced from the glider (blue), ADCP (red), SARAL (green) and J2 (blackgreen) high-rate altimetry data for the 7 individual cases identified in Table 3. Overlapping periods between the different observations are also indicated.

**Specific remarks :**

**Pg. 3, line 1, "Radar altimeters measure the sea surface height (SSH) variations": the sentence is uncorrect. The radar altimeter transmit pulses. The system measures the time pulses take to be reflected back satellite. Time is then converted in distance,corrected for various effects and then referred to earth using orbit altitude (this is the so called SSH). Please be precise**

We agree. It has been reworded for "Radar altimeters allow to estimate the sea surface height (SSH) variations along the satellite tracks at regular interval time".

**Pg. 3, line 2, "ever more accurate observations": more accurate than what? Please clarify**

It has been replaced by "Providing a large number of continuous and accurate observations ..."

**Pg. 3, line 9, "the first global synthetic aperture radar (SAR, or Delay-Doppler) technique": the term "SAR" has to be sued properly to avoid confusion. The reader would understand the first global SAR. SAR, also known as Delay Doppler, is a new coherent processing mode of individual echoes (conventional altimetry uses incoherently processing). Please better clarify**

We agree. It is now : "With the launch of Sentinel-3A, B, in February 2016 and in April 2018, the altimetry constellation was completed by the first instruments always operated at high resolution mode (commonly called Synthetic Aperture Radar or SAR) which increases the along-track data resolution."

**Pg. 3, line 10, "enhanced accuracy and reduced noise": the use of the term "accuracy is wrong". Reduced noise means better precision. The other enhancement is increased native along track resolution**

We have changed this sentence for : "These new altimeters provide enhanced along-track resolution and reduced noise, in comparison to the conventional nadir-looking pulse limited Ku-band instruments used since the beginning of the altimetry era"

**Pg. 3, line 13, "the SWOT a new step forward": reference missing**

A reference has been added : https://swot.jpl.nasa.gov/docs/SWOT_D-79084_v10Y_FINAL_REVA__06082017.pdf

**Pg. 3, lines 14-15, "particularly important to monitor the sea level variations, directly related to our living environments and marine ecosystems": This sentence is too vague.**

**You have to better explain (in term of processes, e.g. flooding, etc.) why sea level changes are more important in the coastal zone than deep waters.**

Done. Now : "In coastal ocean areas, it is particularly important to monitor the sea level variations. About 10% of the world population lives in low-elevation coastal zones (Nicholls and Cazenave, 2010) exposed to hazards such as extreme events, flooding, shoreline erosion and retreat. The latter are expected to increase due to the combined effects of sea level rise, climate change, and increasing human activities. In coastal regions in particular, we expect a lot of advances from modern altimetry instruments and processing techniques."

**Pg. 3, line 16, "from these new altimetry techniques": The statement is vague. Maybe use the term "modern altimetry" to include all technical improvements (Delay Doppler processing, small footprint) as well as reprocessed conventional altimetry (retracking, new/improved corrections, etc.).**

Done. See above.

**Pg. 3, lines 17-18, "The strongest limitation is the modification of the radar echo in the vicinity of land": here the reader might thing that the problem is only land contamination. Indeed, there are other effects , e.g. inhomogeneity of the water surface (Brown altimetry assumes homogeneous scattering). The authors have to be more rigorous here citing proper literature**

This sentence has been replaced by : "The strongest limitation is the modification of the radar echo in the vicinity of land but the sea level estimations derived are also impacted by inhomogeneity in the water surface oberved by the radar and less accurate corrections.".

**Pg. 3, line 27, "new altimetry techniques are intrinsically less sensitive to the land contamination": The statement is not correct. Again the term "techniques" is not appropriate. Moreover, land contamination cannot be used in general sense., e.g. SAR mode has few effect if the track is parallel to land**

This has been reworded for "Moreover, the use of new altimetry techniques provides more robust and accurate measurements closer to the coast and allows to resolve shorter spatial scales (Dufau et al., 2017; Morrow et al., 2017).".

**Pg. 3, line 28, "They provide more robust and accurate measurements, ever closer to the coast": The athors have to provide figures about accuracy as function of distance with concrete examples from bibliography**

We added numbers and references : « Moreover, the use of new altimetry techniques provides more robust and accurate measurements closer to the coast and allows to resolve shorter spatial scales (Dufau et al., 2017; Morrow et al., 2017). As an example, from Birol and Niño (2015), closer than 10 km to the coastline, available SARAL data is still ~60% and only ~31% for Jason-2. From Morrow et al. (2017), in summer SARAL can detect ocean scales down to 35 km wavelength, whereas the higher noise from Jason-2 blocks the observation of scales less than 50-55 km wavelength."

**Pg. 3, lines 29-30, "We can easily predict that the use of altimetry in coastal studies": the reader here expects illustrating major findings from these studies (i.e. state-of-the-art)**

We agree and have completed this sentence. Now : "As a result, the capability of altimetry for the monitoring of coastal ocean dynamics has already been illustrated in a number of studies. Most of them concern shelf and boundary currents (*Bouffard et al.*, 2008; *Birol et. al.*, 2010; *Herbert et al.*, 2011; *Jébri et al.* 2016, 2017). Some others are related to sea level applications (*Cipollini et al.*, 2017a). A more complete review of coastal altimetry applications can be found in *Cipollini et al* (2017b) but we can easily predict that the use of this instrument in coastal studies will be largely extended in the next years."

**Pg. 3, lines 32-33, "Coastal observations are mainly based on in situ instruments and satellite imagery (sea surface temperature and ocean color images): I don't understand this sentence. The coastal observing system is multi-sensor, multi-platform. SAR imagery is especially useful in the coastal zone due to its high spatial resolution. The sentence has to be reworded.**

The sentence has been reworded. Now : « Today, observations used in coastal applications are mainly based on in-situ instruments and satellite imagery (sea surface temperature and ocean color images). »

**Pg. 4, lines 1-2, "in a growing number of regions": Please provide examples**

Done. Now : "In order to answer to the need for monitoring of the coastal ocean environment, in situ observing systems gather informations in a growing number of regions such as along the Australian or US coasts (http://imos.org.au/; https://maracoos.org/operations)."

**Pg. 4, line 2, "in conjunction": better using in "synergy"**

Done.

**Pg. 4, lines 4-5, "in situ observations cover more limited areas and often provide time series that present large gaps.": please provide examples. What means gaps in time series ? are you referring to tide gauges ? buoys? Please better clarify**

It has been clarified. Now: "Nonetheless, in situ observations cover more limited areas and often provide time series that present large gaps which may be several days (buoy data, HF radars) to several months (glider, ship data)."

**Pg. 4, lines 6-7, "satellite imagery is often impacted by clouds and does not provide any direct information on the changes occurring in the water column.": the sentence is confusing the reader. Clouds might be a problem for optical imagery, but not for microwave sensors (e.g. scatterometry, SAR). Moreover, although satellites maps the ocean surface, it si possible to derive info in the water columns (one example is SAR detecting internal waves)**

"Satellite" has been replaced by "optical".

**Pg. 4, line 8, "almost-global synoptic observations". Satellite altimetry provide global coverage. Revisting the same place depends on the mission (e.g. Cryosat is drifting in orbits and revisits the same place in along time). What about "synoptic" ? e.g.Jason takes 10 days to get a global coverage. What do you mean with "almost"? –the sentence in unclear. The reader might be confused, e.g. an optical imagery is synoptic. Infor at all pixels is at same time**

We use the word "synoptic" because satellite altimetry enables to capture a regional general view of the ocean dynamics (in the sense not only local). "Almost-global" is because altimetry observations are not global: they provide observations up to a given latitude.

**Pg. 4, line 11, "conjunction": use "synergy"**

Done.

**Pg. 4, line 13, "ideal": use another word (e.g. laboratory (Béthoux et al., 1999) and explain why in detail)**

The word "ideal" has been replaced by "laboratory". We have also changed the text for "To study the contribution of altimetry amongst other types of coastal ocean measurements, the North-Western Mediterranean Sea (NWMed) represents a laboratory area. First, with a Rossby radius of only ~10 km, the region is associated to a variety of mesoscale and sub-mesoscale dynamical signals (see below). As a result it represents a challenge for altimetry observations. Secondly, the number of in-situ observations is relatively important in this region, allowing comparison with independent data."

**Pg. 4, line 14, "at all time scales"; clarify which scales (i.e. range). →**

We have removed these words (see above).

**Pg. 4, line 20, mesoscale variability is higher in autumn and winter": Why higher in these seasons ? please explain**

Now: "The mesoscale variability is higher in autumn and winter because of the larger baroclinic instability associated to strong and cold winds (*Alberola et al.*, 1995; *Millot*, 1991)."

**Pg. 4, line 25, "to partially capture": please explain why "partially" →**

This sentence has been reworded. Now: "*Birol et al.*, 2010 demonstrated that reprocessed altimetry data are able to capture half of the amplitude of the seasonal NC variability."

**Pg. 4, line 26, "to provide original aspects of the regional circulation": please explain "original aspects"**

This sentence has been reworded. Now: "The altimetry currents have then been used to analyze the regional current variability at seasonal scale."

**Pg. 4, line 27, "coherent circulation patterns": please illustrate these patterns**

This sentence has been reworded. Now: "In the Balearic Sea, the reliability of altimetry currents has been verified by direct comparison with currents derived from gliders and HF radars (*Bouffard et al.,* 2010; *Pascual et al.;* 2015; *Troupin et al.*, 2015). These case studies showed that  altimetry can depict current signals coherent with the other instruments."

**Pg. 4, line 29, "found similarities": which ones?**

This sentence has been reworded. Now: "*Morrow et al.*, 2017 also showed that some of the large scale eddies observed by gliders in the NWMed can be captured by altimetry, in particular by the SARAL mission."

**Pg. 4, line 33, "compared to the other coastal ocean observing systems": please specify which coastal observing systems →**

This sentence is now: "The general objective of this paper is not only to investigate the accuracy of the velocity fields derived from altimetry data next to the coast at different temporal scales , but also to define its contribution from the other coastal ocean observing systems which exist in the region (ship-mounted ADCPs, gliders and HF radars). "

**Pg. 4, line 34, "Ligurian Sea": better to use Ligurian-Provencal basin, because HF radars are not in the Ligurian Sea (as it is usually defined in term of boundaries) →**

Done.

**Pg. 5, line 15, "The performance of SARAL is significantly better than Jason-2": please provide references stating that with figures →**

Now: "The performance of SARAL is significantly better. With a better signal-to-noise ratio, it resolves smaller spatial scales than Jason 2: ~40 km against ~50 km (Dufau et al., 2015, Verron et al., 2018)."

**Pg. 5, line 19, "SARAL tracks 302, 343 and 887": why not using also the other tracks ?**

We chose tracks which are close to in situ datasets and well oriented to capture the NC. The use (and analysis) of all SARAL tracks would not provide more informations in this study and would overload all the figures and discussions.

**Pg. 5, line 23, "Sea Level Anomalies (SLA) every 6-7 km: I am but surprised authors use this low along track sampling (7 km). As the novel aspect is the finer scale of ocean circulation, the authors have to use the high res altimetry (350 m) that are reprocessed using re-tracking.**

Please see the answer to the main comments above. This paragraph has been changed and is now: "For both missions, because it is one of the most often used in coastal altimetry applications, we used first the X-TRACK regional product from the CTOH (doi: 10.6096/CTOH_X-TRACK_2017_02), processed with a coastal-oriented strategy (*Birol et al.*, 2017). It consists in time series of 1-Hz Sea Level Anomalies (SLA) every 6-7 km along the satellite tracks, available from 20/07/2008 to 01/10/2016 for Jason 2 (i.e. 300 cycles) and from 24/03/2013 to 12/06/2016 for SARAL (i.e. 34 cycles). In order to evaluate the skill of

the elementary noisier 20/40-Hz altimetry measurements of the Jason-2/SARAL altimeters for circulation studies, relative to the conventional 1-Hz data, we have also used an experimental high-rate version of these data provided by the CTOH (see section 3.4). The processing is the same than for 1-Hz SLA, except that the high-rate SLA are computed from the 20/40-Hz range data provided in the AVISO L2 products (https://www.aviso.altimetry.fr/fileadmin/documents/data/tools/hdbk_j2.pdf and https://www.aviso.altimetry.fr/fileadmin/documents/data/tools/SARAL_Altika_products_hand book.pdf). The resulting sea level time series are available every ~0.35 km and ~0.17 km along the satellite tracks for Jason-2 and SARAL, respectively. However, we must keep in mind that if the use high-rate altimeter measurements allows to significantly improve the spatial resolution, the resulting SLA are much noisier (see for example *Birol and Delebecque*, 2014). Considering the data availability (see below for the in-situ observations), the study period chosen is 2010-2016 for all altimetry datasets."

**Pg. 6, line 2, "in Morrow et al., 2017, the data located over the continental shelf were discarded": this is further point that support the need of using high res altimetry**

Please see above.

**Pg. 6, lines 1-4, "We did 39 km for the SARAL track 343, 34 km for the SARAL track 887 and 67 km for Jason 2 track 222.. altimetry observations": I see that Morrow et al., 2017 used these figures come from standard along-track data at 1 Hz (7 km) discarding data in the coastal zone. They found some figures about size of structures and optimal filtering. The authors here use reprocessed along-track data at 1 Hz with no retracking applied, but with more data coverage going to the coast. They found lower figures about size of structures. I am confused as we talk about average values and sampling is not changing. Is signal-to-noise better ? you have to demonstrate, because structures can emerge from background noise only the ratio is higher . I think the scale would only change (become finer) only if authors enhance resolution of their altimeter data**

The dataset used does not cover the same period and the same region than in Morrow et al.. In this study all the coastal data were discarded and all the regional spectra were averaged. Here, the computation is done track by track (and then on a mean spectrum computed from much less original samples). The computation of the mesoscale capability is very sensitive to the slope of the mean spectrum analyzed (itself very sensitive to the number of samples and the seasonal situation they capture, the way the data have been post-processed and interpolated to avoid data gaps that complicate the calculation). It explains why our results differ from those of Morrow et al 2017. If we average the results from a larger dataset we would converge to the same numbers. But individual cases may be very different from what is revealed by mean statistics. This is why we have chosen to treat in this paper a wide range of cases: from the temporal mean to individual dates.

**Moreover, the author do not explain why scales vary with close tracks**

The two closest SARAL tracks show almost the same mesoscale capability. The SARAL track 302 and the Jason 2 track 222 are located further so there may be mesoscale processes to take into account. The mesoscale capability corresponds to the scale for which the signal to noise ratio is greater than 1. The Jason 2 data are noisier thus it hides some scales. This explain the differences between SARAL and Jason 2.

**Pg. 6, line 8, "35 km for SARAL": why do you set 35 km if tracks have different scales? Please justify**

We have chosen to be consistent in the processing of the SARAL data for the different tracks in order to facilitate the comparison and discussion. Moreover, the mesoscale capability deduced from a spectral approach is a statistical result and corresponds to a mean situation (which may be biased by the number of samples analyzed and the way spectra have been computed). In some cases we may be able to observe smaller structures than expected and in some others noise will clearly remain (as shown in the individual cases). The choice of the optimal cutoff  frequency does not seem obvious at all. Anyway, this section has been slightly changed to clarify. Now: "In order to have the best signal-over-noise ratio, we then filtered the data with a low-pass Loess filter, using a cut-off frequency of 35 km for SARAL. Note that we have chosen a single value for the different SARAL tracks in order to have the same data processing and facilitate the comparison between the different datasets. For Jason-2, we chose the option of using a processing as close as possible from the one of SARAL and then used a cut-off frequency of 40 km. The same low-pass filters were used for both 1-Hz and high-rate SLAs. One need then to keep in mind that noise remains in the filtered Jason 2 data."

**Pg. 6, line 13, "from (Rio et al.,2014)": change to "Rio et al., (2014)" Moreover, the authors have to demonstrate that this MDT is accurate going closer to the coast as in open ocean (this product was not generated to be used in the coastal zone)**

We agree. We chose to work with the regional MDT from Rio et al., 2014 which was validated against in situ datasets. Compared to the previous MDT from Rio et al., 2007, it has a better resolution (1/16 degree vs 1/8 °) and the regional circulation is better resolved (see Rio et al, 2014 but we have also done our own diagnostics).  We have added the following sentences:

"The MDT product used is a regional product with an horizontal resolution of 1/16° (lower than the altimetry resolution in the along-track direction). Compared to other products, it allows a better representation of the NC in the Ligurian Sea (Rio et al., 2014)."

"(Rio et al., 2014)" has been changed for "Rio et al., (2014)".

**Pg. 6, lines 29-30 "The ones being too short or moving too far away from an average trajectory": please be rigorous in stating "too short" and "too far away**

Right. Now: "The ones being too short (<60 km) or moving too far away (>15 km) from an average trajectory computed from the individual ones were discarded. "

**Pg. 7, line 6, "points and the was too": typo to correct**

Corrected. We added the word "horizontal" in the text.

**Pg. 6, lines 12-13, "compare the currents derived from these data with the currents measured or derived from the other instruments": becareful that altimeter derived currents from altimetry are not equivalent to currents measured e.g. from ADCPs**

We agree. Now: "In order to estimate currents as close as possible to the currents measured or derived from the other instruments (see below), ...".

**Pg. 8, line 17, "HF radar is roughly 60x40": is this the area covered? How much is accuracy of currents? Please discuss bibliography**

The HF radars section has been developed. Now : "The HF radars data used here, which are also part of the MOOSE network (Zakardjian and Quentin, 2018), targets the area off the coast of Toulon as a key zone conditioning the behaviour of the North Current just upstream of the Gulf of Lions due to a sharp bathymetry and several islands thet deviate a stronger NC southwestward, significant cross-shelf exchanges correlated to the strongnorth-westerlies present in the region (Mistral, Tramontane) as well as a marked mesoscale variability of the NC (e.g., Guihou et al., 2013). The system consists in two HF (16 MHz) Wellen Radar (WERA) instruments installed near Toulon in monostatic (Cap Sicié station) and bistatic (Cap Bénat/Proquerolles island stations) 8-antenna configurations (see Quentin et al., 2013, 2014 for details). The systems are working with a 50 kHz bandwidth, resulting in a 3 km range resolution, a direction finding method based on MUSIC (Lipa et al., 2006, Molcard et al., 2009) allowing a 2 degree azimuthal resolution and with a time integration of 20 minutes. The radial velocities maps are means over a 1 hour time window and cartesian total velocities are then reconstructed on a regular 2 x 2 km grid. An assesment of this HF Radar site can be found in Sentchev et al. (2017) who found an overall good agreement between derived radial velocities and in situ ADCP, with relative errors of 1 and 9 % and root mean square (RMS) differences of 0.02 and 0.04 m/s, slightly increased, in velocity and direction, for the reconstructed total velocities, but mainly in conditions of unstationnary wind forcing. The MOOSE HF radar data base used here is made of daily (one diurnal lunar period of 25h), averaged surface currents computed from the re-processed hourly total velocity data (QC level L3B, i.e., velocity threshold and Geometric Dilution of Precision – GDOP - tests passed) with additional cleaning of residual RFI outliers using outlier-removal algorithm based on the number of L3B valid data, variance and mean over a one intertial period window (17h at 43°N). The data are hence hence tides and inertial oscillation filtered. The time series starts in May 2012 and ends in September 2014 with a total of 732 days of available data. The size of the area covered by total velocities after the GDOP test is roughly 60x40 km and it is located about 170 kmwestward of the glider and ADCP observations (as well as of the altimetry tracks we have chosen to focus on in this study)."

**Pg. 9, lines 4-5, "to altimetry data still remains limited in the 10- 15 km coastal band" The statement is wrong. With reprocessed high res altimetry tat adopts retracking one can go closer to the coast.**

We changed into "to 1Hz-altimetry data". However, if more altimetry data are now available thanks to efforts done in retracking and reprocessing, 1) their uncertainty is larger (and remain unknown) and 2) they are still available only for some expert users (because only experimental data sets exist, usually at level 2 or level 2P, covering only one or a few missions, or a few areas or a few years). So we really believe that for most users altimetry data still remains limited in the 10- 15 km coastal band.

**Pg. 9, line 19, "HF radars and altimeters observe the ocean surface": altimetry provides geostrophic currents that are derived from SLA where tides and atmospheric effects (wind and pressure) are removed. HF radar provides the real total current at surface. . Also gliders provide only the baroclinic component of currents. ADCP measure currents at differet layers. Authors have to discuss in detail these differences.**

The differences are explained in details in the the next section ("physical content") and in in the data description.

**Pg. 9, lines 33-34, "the gliders and altimeters are clearly the closest in terms of current information derived.". I don't agree. Gliders miss the barotropic component due to atmospheric effects that in the coastal zone is not negligible**

Part of the barotropic component is also removed from altimetry through the DAC correction (e.g. the barotropic response to the HF wind fluctuations). Moreover, in the glider data, an estimate of depth-average currents was added to the velocity data as an estimation of the barotropic component. So we really think that the question of the details of the differences between altimetry and glider is a complex issue and goes beyond the scope of this paper. We think that in this region ageostrophic motions are very important and that among all the platforms analyzed in this study, gliders and altimetry are the closest in terms of current information derived. However we agree that we needed to be clearer in the text and made a number of changes in section 2.2.d "physical content" accordingly. Now:

"Moreover, the different instruments do not capture the same physical content. The ADCP and the HF radars measure both the total instantaneous velocities when the gliders and altimeters allow to derive only the geostrophic current component perpendicular to the satellite or glider track (i.e. excluding the ageostrophic parts such as wind-driven surface current, tidal currents, internal waves, etc…, and the current component parallel to the track). Unlike the other current data sources used here, altimetry gives only access to current anomalies. But the addition of a synthetic MDT allows to overcome this difficulty if its quality is good enough to derive a reliable mean velocity field. After the addition of the MDT, the gliders and altimeters are clearly the closest in terms of current information derived. However, the glider currents are computed from hydrographic measurement profiles with a reference level of 500 m. They miss the barotropic and the deeper baroclinic geostrophic current componentss when altimetry and MDT allow to estimate absolute geostrophic currents representative of the horizontal density gradients integrated over the whole water column. In this study, in order to minimize (as far as possible) the differences between the current data sets, we performed a projection of the ADCP velocities to obtain the current component perpendicular to the ship transects.

Concerning the gliders , estimates of depth-average currents computed following *Testor et al.*, 2018 approach were added to the velocity data as an estimation of the barotropic component."

**Pg. 10, line 22, "from March 2013 to October 2014": I am not sure this is a good approach. Mean flows have sense if you average by month, season or annual.**

We agree but the period which is common for all the types of observations is really limited and we have chosen to have a maximum of samples to compute the statistics. Our objective here is to quantify the differences between the currents derived from the different platforms and not to discuss the seasonal or annual variability.

**Pg. 11, lines 1-2, "Fig. 2, where one can notice a very good consistency of the mean currents derived from all the different instruments."**

**Pg. 11, lines 32-33, "HF radars (∼-0.44 m/s) than in altimetry (∼-0.29 m/s)": Why this difference ? please explain.**

Right. We added the following sentence: "This difference is probably due to the ageostrophic motions captured by the HF radars but not by altimetry, and to the differences in the data resolution."

**Pg. 12, line 1, "we observe values of 0.12-0.2 m/s": Mean values are around 0.3 m/s and variability is of same order of magnitude (more or less). Is this picture confirmed by bibliography ?**

Very few NC studies use altimetry data but in Birol et al., 2010, values between 0.1 and 0.2 m/s were found. It is due to the too low resolution of the data that capture only one part of the NC variability (explained below in the text). From this work we demonstrate that 1-Hz altimetry data (corresponding to the standard products available for the scientific community) are able to capture 50% of the variability of the geostrophic NC component.

[revised manuscript text omitted]

Birol, F., N. Fuller, F. Lyard, M. Cancet, F. Niño, C. Delebecque, S. Fleury, F. Toublanc, A. Melet, M. Saraceno, and F. Léger. 2017. Coastal applications from nadir altimetry: Example of the X-TRACK regional products. *Advances in Space Research*, 59(4):936–953, https://doi.org/10.1016/j.asr.2016.11.005.

Bonnefond, P., J. Verron, J. Aublanc, K. N. Babu, M. Bergé-Nguyen, M. Cancet, A. Chaudhary, J. F. Créteaux, F. Frappart, B. J. Haines, O. Laurain, A. Ollivier, J. C. Poisson, P. Prandi R. Sharma, P. Thibaut, C. Watson. 2018. The Benefits of the Ka-Band as Evidenced from the SARAL/AltiKa Altimetric Mission: Quality Assessment and Unique Characteristics of AltiKa Data. *Remote Sensing* 10 (83) , https://doi:10.3390/rs10010083

Bosse A., P. Testor, N. Mayot, L. Prieur, F. D'Ortenzio, L. Mortier, H. Le Goff, C. Gourcuff, L. Coppola, H. Lavigne, P. Raimbault. 2017. A submesoscale coherent vortex in the Ligurian Sea: From dynamical barriers to biological implications. *Journal of Geophysical Research: Oceans* 122(8) 6196-6217, https://doi:10.1002/2016JC012634

Bosse, A., P. Testor, L. Mortier, L. Prieur, V. Taillandier, F. d'Ortenzio, and L. Coppola. 2015. Spreading of Levantine Intermediate Waters by submesoscale coherent vortices in the northwestern Mediterranean Sea as observed with gliders. *Journal of Geophysical Research: Oceans*, 120(3):1599–1622, https://doi.org/10.1002/2014JC010263.

Bouffard, J., A. Pascual, S. Ruiz, Y. Faugère, and J. Tintoré. 2010. Coastal and mesoscale dynamics characterization using altimetry and gliders: A case study in the Balearic Sea. *Journal of Geophysical Research: Oceans*, 115(C10):C10029, https://doi.org/10.1029/2009JC006087.

Bouffard J., S. Vignudelli, P. Cipollini, Y. Menard. 2008. Exploiting the potential of an improved multimission altimetric data set over the coastal ocean. *Geophysical Research Letters*, 35(10). 10.1029/2008GL033488

Casella, E., A. Molcard, and A. Provenzale. 2011. Mesoscale vortices in the Ligurian Sea and their effect on coastal upwelling processes. *Journal of Marine Systems*, 88(1):12–19, https://doi.org/10.1016/j.jmarsys.2011.02.019.

Cipollini P., F. M. Calafat, S. Jevrejeva, A. Melet, P. Prandi. 2017. Monitoring Sea Level in the Coastal Zone with Satellite Altimetry and Tide Gauges. *Surveys in Geophysics* 38(1), 33-57

Cipollini P., J. Benveniste, F. Birol, J. Fernandes, E. Obligis, M. Passaro, P.T. Strub, P. Thibaut, G. Valladeau, S. Vignudelli and J. Wilkin, 2017b. Satellite altimetry in coastal regions. CRC Book, Chapter 12.

Crépon M., L. Wald, J. M. Monget. 1982. Low-frequency waves in the Ligurian Sea during December 1977. *Journal of Geophysical Research*. 87(C1) 595-600. 10.1029/JC087iC01p00595

Dufau C., M. Orsztynowicz, G. Dibarboure, R. Morrow, P. Y. Le Traon. 2016. Mesoscale resolution capability of altimetry: Present and future. *Journal of Geophysical Research: Oceans*, 121(7) 4910-4927 https://doi.org/10.1002/2015JC010904

Durand, F., F. Marin, J.-L. Fuda, and T. Terre. 2017. The East Caledonian Current: A Case Example for the Intercomparison between AltiKa and In Situ Measurements in a Boundary Current. *Marine Geodesy*, 40(1):1–22, https://doi.org/10.1080/01490419.2016.1258375.

Font J., E. Garcialadona, E. G. Gorriz. 1995. The seasonality of mesoscale motion in the northern current of the western Mediterranean – several years of evidence. Oceanologica Acta 18(2) 207-219

Gòmez-Enri, J., Cipollini, P., Passaro, M., Vignudelli, S., Tejedor, B., & Coca, J. 2016 Coastal Altimetry Products in the Strait of Gibraltar. IEEE Transactions on Geoscience and Remote Sensing, 54 (9), 5455-5466.

Gommenginger C., P. Thibaut, L Fenoglio-Marc, G. Quartly, X. Deng, J. Gomez-Enri, P. Challenor, Y. Gao. 2011. Retracking Altimeter Waveforms Near the Coasts. *Coastal Altimetry* 61-101

Guihou, K., J. Marmain, Y. Ourmières, A. Molcard, B. Zakardjian, and P. Forget. 2013. A case study of the mesoscale dynamics in the North-Western Mediterranean Sea: a combined data–model approach. *Ocean Dynamics*, 63(7):793–808, **https://doi.org/10.1007/s10236-013-0619-z**.

Herbert, G., N. Ayoub, P. Marsaleix, F. Lyard. 2011. Signature of the coastal circulation variability in altimetric data in the southern Bay of Biscay during winter and fall 2004. Journal of Marine Systems, 88 (2), 139-158, https://doi.org/10.1016/j.jmarsys.2011.03.004

Jébri F., F. Birol, B. Zakardjian, J. Bouffard, C. Sammari. 2016. Exploiting coastal altimetry to improve the surface circulation scheme over the central Mediterranean Sea. *Journal of Geophysical Research: Oceans* 121(7) 4888-4909, 10.1002/2016JC011961

Jébri F., B. Zakardjian, F. Birol, J. Bouffard, L. Jullion, C. Sammari. 2017. Interannual Variations of Surface Currents and Transports in the Sicily Channel Derived From Coastal Altimetry. *Journal of Geophysical Research: Oceans* 122(11) 8330-8353. 10.1002/2017JC012836

Lipa, B, Nyden, B. Ulman, D. S., & Terrill E. 2006. SeaSonde Radial Velocities: Derivation and Internal Consistency, IEEE Journal of Oceanic Engineering, vol. 31, 850-861.

Liu, Y., H. Kerkering, and R.H., Weisberg (Editors) 2015, Coastal Ocean Observing Systems, 461 PP., ISBN 978-0-12-802022-7, Elsevier (Academic Press), London, UK.

Millot, C. 1991. Mesoscale and seasonal variabilities of the circulation in the western Mediterranean. *Dynamics of Atmospheres and Oceans*, 15(3):179–214, **https://doi.org/10.1016/0377-0265(91)90020-G**.

Molcard A., Poulain P.M., Forget P., Griffa A., Barbin Y., Gaggelli J., De Maistre J.C., Rixen M., 2009. Comparison between VHF radar observations and data from drifter clusters in the Gulf of La Spezia (Mediterranean Sea), J. Marine Systems 78 S79-S89.

Morrow, R., and P.-Y. Le Traon. 2012. Recent advances in observing mesoscale ocean dynamics with satellite altimetry. *Advances in Space Research*, 50(8):1062–1076, https://doi.org/10.1016/j.asr.2011.09.033.

Morrow, R., A. Carret, F. Birol, F. Nino, G. Valladeau, F. Boy, C. Bachelier, and B. Zakardjian. 2017. Observability of fine-scale ocean dynamics in the northwestern Mediterranean Sea. *Ocean Science*, 13(1):13–29, https://doi.org/10.5194/os-13-13-2017.

Niewiadomska, K., H. Claustre, L. Prieur, and F. d'Ortenzio. 2008. Submesoscale physical-biogeochemical coupling across the Ligurian current (northwestern Mediterranean) using a bio-optical glider. *Limnology and Oceanography*, 53(5):2210,.

Ourmières, Y., B. Zakardjian, K. Béranger, and C. Langlais. 2011. Assessment of a NEMO-based downscaling experiment for the North-Western Mediterranean region: Impacts on the Northern Current and comparison with ADCP data and altimetry products. *Ocean Modelling*, 39(3–4):386–404, https://doi.org/10.1016/j.ocemod.2011.06.002.

Pascual, A., A. Lana, C. Troupin, S. Ruiz, Y. Faugère, R. Escudier, and J. Tintoré. 2015. Assessing SARAL/AltiKa Data in the Coastal Zone: Comparisons with HF Radar Observations. *Marine Geodesy*, 38(sup1):260–276, https://doi.org/10.1080/01490419.2015.1019656.

Pascual A. and D. Gomis. 2003. Use of Surface Data to Estimate Geostrophic Transport. *Journal of Atmospheric and Oceanic Technology* 20, 912-926

Passaro M., P. Cipollini, S. Vignudelli, G. D. Quartly, H. M. Snaith. 2014. ALES: A multi-mission adaptive subwaveform retracker for coastal and open ocean altimetry. Remote Sensing of Environment, 145, 173-189

Passaro, M., S. Dinardo, G. D. Quartly, H. M. Snaith, J. Benveniste, P. Cipollini, B. Lucas. 2016. Cross-calibrating ALES Envisat and CryoSat-2 Delay–Doppler: A coastal altimetry study in the

Indonesian Seas. *Advances in Space Research* 58(3), 289-303, http://dx.doi.org/10.1016/j.asr.2016.04.011

Petrenko A., C. Dufau and C. Estournel, 2008. Barotropic eastward currents in the western Gulf of Lion, north-western Mediterranean Sea, during stratified conditions. *Journal of Marine Systems*, 74(1-2), 406-428.

Piterbarg, L., V. Taillandier, and A. Griffa. 2014. Investigating frontal variability from repeated glider transects in the Ligurian Current (North West Mediterranean Sea). *Journal of Marine Systems*, 129:381–395, **https://doi.org/10.1016/j.jmarsys.2013.08.003**.

Quentin C., Y. Barbin, L. Bellomo, P. Forget, J Gagelli., S. Grosdidier, C.-A. Guerin, K Guihou., J. Marmain, A. Molcard, B. Zakardjian, P. Guterman, K. Bernardet, 2013. HF radar in French Mediterranean Sea: an element of MOOSE Mediterranean Ocean Observing System on Environment. OCOSS'2013 Proceedings, vol. p.25-30, hal-00906439

Quentin C., Y. Barbin, L. Bellomo, P. Forget, D. Lallarino, J. Marmain, A. Molcard, B. Zakardjian, P.,Guterman, K. Bernardet, 2014. High Frequency Surface Wave Radar in the French Mediterranean Sea: an element of the Mediterranean Ocean Observing System for the Environment. In Buch E, Antoniou Y, Eparkhina D, Nolan G (Eds.), Operational Oceanography for Sustainable Blue Growth. Proceedings of the Seventh EuroGOOS International Conference. 28-30 October 2014, Lisbon, Portugal. ISBN 978-2-9601883-1-8

Rio, M.-H., A. Pascual, P.-M. Poulain, M. Menna, B. Barceló, and J. Tintoré. 2014. Computation of a new mean dynamic topography for the Mediterranean Sea from model outputs, altimeter measurements and oceanographic in situ data. *Ocean Sciences*, 10(4):731–744, https://doi.org/10.5194/os-10-731-2014.

Roblou L., J. Lamouroux, J. Bouffard, F. Lyard, M. Le Hénaff, A. Lombard, P. Marsaleix, P. De Mey, F. Birol. 2011. Post-processing altimeter data towards coastal applications and integration into coastal models. *Coastal Altimetry*, 217-246

Sammari, C., C. Millot, and L. Prieur. 1995. Aspects of Seasonal and Mesoscale Variabilities of the Northern Current in the Western Mediterranean-Sea Inferred from the PROLIG-2 and PROS-6 Experiments. *Deep-Sea Research Part I-Oceanographic Research Papers*, 42(6):893–917, **https://doi.org/10.1016/0967-0637(95)00031-Z**.

Sentchev A., Forget P., Fraunie P., 2017. Surface current dynamics under sea breeze conditions observed by simultaneous HF radar, ADCP and drifter measurements.

Taupier-Letage, I., and C. Millot. 1986. General hydrodynamical features in the Ligurian Sea inferred from the DYOME experiment. *Oceanologica Acta*, 9(2):119–131,.

Testor, P., A. Bosse, L. Houpert, F. Margirier, L. Mortier, H. Legoff, D. Dausse, M. Labaste, J. Karstensen, D. Hayes, A. Olita, A. Ribotti, K. Schroeder, J. Chiggiato, R. Onken, E. Heslop, B. Mourre, F. d'Ortenzio, N. Mayot, H. Lavigne, O. de Fommervault, L. Coppola, L. Prieur, V. Taillandier, XD. De Madron, F. Bourrin, G. Many, P. Damien, C. Estournel, P. Marsaleix, I. Taupier-Letage, P. Raimbault, R. Waldman, M. N. Bouin, H. Giordani, G. Caniaux, S. Somot, V. Ducrocq, P. Conan (2018). Multiscale observations of deep convection in the northwestern Mediterranean Sea

during winter 2012–2013 using multiple platforms. *Journal of Geophysical Research: Oceans*, 123(3) 1745-1776. https://doi.org/10.1002/2016JC012671

Troupin, C., A. Pascual, G. Valladeau, I. Pujol, A. Lana, E. Heslop, S. Ruiz, M. Torner, N. Picot, and J. Tintoré. 2015. Illustration of the emerging capabilities of SARAL/AltiKa in the coastal zone using a multi-platform approach. *Advances in Space Research*, 55(1):51–59, https://doi.org/10.1016/j.asr.2014.09.011.

Valladeau, G., P. Thibaut, B. Picard, J. C. Poisson, N. Tran, N. Picot, A. Guillot. 2015. Using SARAL/AltiKa to improve Ka-band altimeter measurements for coastal zones, hydrology and ice: The PEACHI prototype. *Marine Geodesy*, 38(sup1), 124-142, http://dx.doi.org/10.1080/01490419.2015.1020176.

Verron, J., P. Bonnefond, L. Aouf, F. Birol, S. A. Bhowmick, S. Calmant, T. Conchy, J.-F. Crétaux, G. Dibarboure, A. K. Dubey, Y. Faugère, K. Guerreiro, P. K. Gupta, M. Hamon, F. Jebri, R. Kumar, R. Morrow, A. Pascual, M. I. Pujol, E. Rémy, F. Rémy, W. H. F. Smith, J. Tournadre, and O. Vergara. 2018. The benefits of the Ka-band as evidenced from SARAL/AltiKa altimetric mission. Part 2: scientific applications. 2018. *Remote Sensing* 10(2), 163 https://doi.org/10.3390/rs10020163

Xu, Y., and L.-L. Fu. 2012. The Effects of Altimeter Instrument Noise on the Estimation of the Wavenumber Spectrum of Sea Surface Height. *Journal of Physical Oceanography*, 42(12):2229–2233, https://doi.org/10.1175/JPO-D-12-0106.1.

[revised manuscript text omitted]

---

## Author Response (AR2)

Alice Carret
Laboratoire d'Etudes en Géophysique et Océanographie Spatiales (LEGOS)
OMP
14 Avenue Edouard Belin
31400 Toulouse
France
Tel : 00-33-5-61-33-26-81
Email : alice.carret@legos.obs-mip.fr

Subject : Manuscript os-2018-76 revision

19 February, 2019

Ocean Science

Dear Editor,

Please find enclosed the revised version of our manuscript entitled "Synergy between in situ and altimetry data to observe and study the Northern Current variations (NW Mediterranean Sea)" , by A. Carret and co-authors.

We thank the editor and the reviewers for their suggestions for revision and their technical corrections. In the following pages we detail the answer to the comments made by the reviewers. We have also included a marked-up manuscript version.

We hope that our answers and corrections will be satisfying and that our manuscript is now suitable for publication.

Best regards,

Alice Carret and co-authors.

**It is still not very clear what is original in the Discussion section. In the article there are many new results, but in the discussion section there are not highlighted properly.**

The conclusion has been significantly rewritten in order to clarify the major findings of this study. It is now :

[revised manuscript text omitted]

**Unnecessary parenthesis are still present in the manuscript. In particular from the second half to the end.**

Right. We have removed a large number of parenthesis throughout the whole text.

**Abstract: replace "Satellite altimetry obviously … []" by "Thus, satellite altimetry…[]"**

It has been replaced.

**Introduction:**

- **"are much more difficult to interpret …[]" than what?**

We have added "than in the open ocean".

- **Last line: number of section missing**

It has been corrected.

**Section 2.1:**

- **Replace "elementary"**

We have removed "elementary" because, obviously, it does not bring any information in the sentence. We have also removed the word noisier because it is redundant with what is written after on the resulting SLA.

**Section 2.2a: missing word in second paragraph ("and the was too strong")**

Right. Now: "In practice data were discarded when the latitude was not monotically varying or when the angle between 2 consecutive points and the mean direction of the reference transect was too strong".

**Section 2.2b: last sentence: as written, the reader might ask why then you don't choose a deeper level.**

Now: "In this study, we focused on the 34 m-depth cell, in order to strongly reduce the surface instrumental errors.".

**Section 3:**

- **You could delete the first part of the sentence: "Note first that all the"**

Done.

- **"NC current" no need to repeat "current" ?**

Right. We have removed the word "current" after "NC" throughout the text.

**Section 3.2:**

- **"seasonality of the difference" what is that?**

This sentence has been reworded. Now: "Two physical processes can explain that the differences between the different types of current measurements vary as a function of the season".

- **Last sentence: please remove parenthesis. And replace "good result"**

We have removed the parenthesis. This sentence has been rephrased. Now: "Despite the spatial resolution of altimetry data and the width and very coastal location of the NC current, the amplitude of its seasonal variations captured by the Jason2 track 222 along the French coast is 55-60% of the amplitude captured by both the gliders and ADCPs."

**Section 3.3:**

- **Last paragraph: "seasonal tendency" what is that? Trends are not studied here.**

"Seasonal tendency" has been replaced by "seasonal average".

**Section 3.4:**

- **Second sentence: do not start with "but"**

  It has been replaced by "However".

- **Second paragraph: what is "NC hydrodynamics" ?**

  "NC hydrodynamics" has been replaced by "NC variations".

- **Last paragraph: parenthesis, "but"**

  We have rephrased this sentence. Now: "However the resulting current fields depend crucially on the strategy followed for data processing, -including retracking and corrections-, screening and filtering."

- **There is a second Section 3.4. Check numbering of Sections. Unnecessary parenthesis everywhere, "but" at the beginning of several sentences again, and strange symbols.**

  Right. We have corrected the numbering of the sections, removed parenthesis, changed "but" into "however" and removed the strange symbols.

**Section 4.**

- **First paragraph: incomplete sentence**

  Right. We completed the sentence. It is now : "In this study, the systematic comparison of the current data derived from the different platforms provide new insights into the biases that their differences cause in the estimations of the NC characteristics."

- **Second paragraph: Inertial oscillations are not "unsteady ageostrophic currents". Please verify.**

  We do not agree with the reviewer. Inertial oscillations are a non-stationary motion and then unsteady ageostrophic currents.

[revised manuscript text omitted]